# Haploinsufficiency of *Trp53* dramatically extends the lifespan of Sirt6-deficient mice

Shrestha Ghosh[1,2], Sheung Kin Wong[1,2], Zhixin Jiang[1,2], Baohua Liu[3], Yi Wang[1], Quan Hao[1], Vera Gorbunova[4], Xinguang Liu[5], Zhongjun Zhou[1,2]*

[1]School of Biomedical Sciences, LKS Faculty of Medicine, The University of Hong Kong, Pok Fu Lam, Hong Kong; [2]Shenzhen Institute of Research and Innovation, The University of Hong Kong, Pok Fu Lam, Hong Kong; [3]School of Medicine, Shenzhen University, Shenzhen, China; [4]Rochester Aging Research Center, University of Rochester, Rochester, United States; [5]Institute for Aging Research, Guangdong Provincial Key Laboratory of Medical Molecular Diagnostics, Guangdong Medical University, Dongguan, China

**Abstract** Mammalian sirtuin 6 (Sirt6) is a conserved $NAD^+$-dependent deacylase and mono-ADP ribosylase that is known to be involved in DNA damage repair, metabolic homeostasis, inflammation, tumorigenesis, and aging. Loss of *Sirt6* in mice results in accelerated aging and premature death within a month. Here, we show that haploinsufficiency (*i.e.*, heterozygous deletion) of *Trp53* dramatically extends the lifespan of both female and male *Sirt6*-deficient mice. Haploinsufficiency of *Trp53* in *Sirt6*-deficient mice rescues several age-related phenotypes of Sirt6-deficient mice, including reduced body size and weight, lordokyphosis, colitis, premature senescence, apoptosis, and bone marrow stem cell decline. Mechanistically, SIRT6 deacetylates p53 at lysine 381 to negatively regulate the stability and activity of p53. These findings establish that elevated p53 activity contributes significantly to accelerated aging in Sirt6-deficient mice. Our study demonstrates that p53 is a substrate of SIRT6, and highlights the importance of SIRT6-p53 axis in the regulation of aging.
DOI: https://doi.org/10.7554/eLife.32127.001

*For correspondence:
zhongjun@hku.hk

**Competing interests:** The authors declare that no competing interests exist.

## Introduction

Mammalian sirtuin 6 (Sirt6), a conserved $NAD^+$-dependent deacylase and mono-ADP ribosylase, has been implicated in a range of regulatory pathways involved in premature aging and other age-associated pathologies (*Kugel and Mostoslavsky, 2014*). Mice deficient in *Sirt6* exhibit severe premature aging phenotypes, including metabolic defects, lymphopenia, osteoporosis, genomic instability, and early death by 4 weeks of age (*Mostoslavsky et al., 2006*). Additionally, the ectopic expression of Sirt6 has been shown to extend the lifespan of male mice by 15% (*Kanfi et al., 2012*). Sirt6 was first identified to deacetylate histone H3 at lysine 9 and 56 (in a $NAD^+$-dependent manner) at the telomeres, promoter regions of target genes and globally as well, to mediate DNA damage repair, maintain telomeric metabolism, suppress NF-κB pathway in promoting longevity and regulate cell cycle (*Kawahara et al., 2009*; *Michishita et al., 2008*; *Michishita et al., 2009*; *Yang et al., 2009*). The intricate roles of Sirt6 in DNA damage repair and metabolic regulation have been reported to stem from its inherent $NAD^+$-dependent deacetylating activity on histones, CtIP, GCN5 and others, apart from its NAD+-dependent mono-ADP ribosylase activity (*Dominy et al., 2012*; *Kaidi et al., 2010*; *Mao et al., 2011*). These and other foundational studies have established the importance of Sirt6 in the regulation of processes contributing to aging and longevity. Although Sirt6-mediated

**eLife digest** Almost without exception, mammals age as they grow older. Older mammals are at greater risk of diseases like cancer, and have fewer stem cells that would otherwise help to keep their organs healthy. Some of the proteins that regulate and impact upon aging have been identified. One of these is an enzyme called SIRT6, which is thought to promote longevity. Mice without any SIRT6 suffer from severe premature aging, and rather than living up to two years like normal mice, SIRT6-deficient mice die within one month of their birth. The mutant mice also lose stem cells and exhibit signs of organ degeneration and body wasting.

Another protein called p53 is well known for having the opposite effect to SIRT6: it accelerates aging and helps to prevent tumor growth. However, it was unclear if p53 is also involved in the processes that lead to the premature death of mice without SIRT6.

Now, Ghosh et al. report that mouse cells and tissues without SIRT6 have more p53 compared to control samples. Biochemical experiments showed that the SIRT6 and p53 proteins physically interact, and that SIRT6 could use its enzymatic activity to remove a chemical modification, called acetylation, from p53. Without this specific acetylation, p53 became less stable and its levels dropped. Consequently, p53 was stabilized in the SIRT6-deficient cells. When Ghosh et al. deleted one copy of the gene that codes for p53 in SIRT6-deficient mice, mutant mice that had before only lived for a month now lived for up to sixteen months. Additionally, the mice were healthier, showing fewer signs of aging: for example, they had more immune cells and stem cells, straighter spines, and showed less gut inflammation and less body wasting.

These findings suggest that SIRT6 does indeed inhibit p53 to counteract the normal aging process. Future experiments may explore if this regulation also holds true in human cells. Detailed knowledge of these molecular interactions could also open up more research into therapies against cancer and aging.

DOI: https://doi.org/10.7554/eLife.32127.002

regulation has been reported for several signaling pathways (for example, NF-κB, AKT, and IGF1) (*Kanfi et al., 2012*; *Kawahara et al., 2009*; *Pan et al., 2016*; *Xiao et al., 2010*), the key mechanism underlying the severe acceleration of aging and premature death in Sirt6-deficient mice remains elusive.

*Trp53*, also denoted as *p53*, is a tumor suppressor gene that triggers cell cycle arrest, apoptosis, and/or senescence (*Rufini et al., 2013*; *Vousden and Prives, 2009*). p53 has been widely implicated in premature senescence and aging (*Hinkal et al., 2009*; *Poyurovsky and Prives, 2010*; *Varela et al., 2005*). p53 was the first non-histone protein identified to undergo acetylation as a post-translational modification (*Gu and Roeder, 1997*). Ever since, a range of lysine residues have been identified across the domains of p53, with majority of acetylated lysine residues being concentrated in the C-terminus region of p53 (*Brooks and Gu, 2011*; *Gu and Zhu, 2012*; *Gu and Roeder, 1997*; *Kruse and Gu, 2009*). Lysine 382, localized in the C-terminus of p53, is identified to undergo marked acetylation followed by lysine 381 and 373 (*Gu and Roeder, 1997*). Given the stunted half-life of around 30 min for p53, these acetylations have been reported to impart stability to p53, which then mediates a range of downstream functions in response to DNA damage, oncogenic stress, age-associated abnormalities, tumor suppression and others (*Donehower, 2009*; *Lozano, 2010*; *Vousden and Prives, 2009*).

In laminopathy-based progeria, mutant lamin A results in elevated p53 signaling and reducing p53 level significantly ameliorates premature aging phenotypes and extends lifespan (*Varela et al., 2005*), thus suggesting that activation of p53-dependent downstream signaling accelerates aging. Interestingly, lamin A is an endogenous activator of SIRT6 and mutant lamin A results in compromised SIRT6 deacetylase activity (*Ghosh et al., 2015*). Given that functional defects in Sirt6 and activation of p53 are both associated with laminopathy-based premature aging, it is conceivable to speculate that loss of Sirt6 may also impact p53 signaling. To determine whether aberrant p53 signaling contributes to the severe progeroid phenotypes in Sirt6-deficient mice, we generated Sirt6-deficient mice with haploinsufficiency (i.e., heterozygous deletion) of *p53* and monitored the growth and survival of the compound mutant mice along with their wild-type and Sirt6 single knockout (KO)

littermates. We assessed and compared a range of premature aging-associated abnormalities in our generated compound mutant mice (Sirt6$^{-/-}$Trp53$^{+/-}$ mice), which have been previously reported in Sirt6 single KO (Sirt6$^{-/-}$Trp53$^{+/+}$) mice (**Mostoslavsky et al., 2006**). Our findings, revealing the direct link between Sirt6 and p53, underscore the critical role of the Sirt6-p53 regulatory axis in aging and age-associated diseases, which has potential therapeutic implications in healthy aging management.

## Results

### Upregulation of p53 signaling upon deficiency of Sirt6

To determine whether p53 signaling is involved in the causation of premature senescence upon loss of Sirt6, we first analyzed the expression of some classical downstream targets of p53 in Sirt6-proficient (Sirt6$^{+/+}$Trp53$^{+/+}$) and deficient (Sirt6$^{-/-}$Trp53$^{+/+}$) primary mouse embryonic fibroblasts (MEFs) obtained from littermates, and explored the possibility that functional cross-talk may occur between these two proteins. To this end, 3 batches of primary MEF cells collected from independent batches of embryos were analyzed. To avoid mutations in p53 pathway occurring spontaneously over serial passaging of MEFs, only primary MEFs of P1 or P2 passage were used in experiments, unless stated otherwise. Intriguingly, a range of downstream targets of p53, including p21, Puma, Noxa, Bax and Ddit4 (**Poyurovsky and Prives, 2010**), were upregulated in Sirt6$^{-/-}$Trp53$^{+/+}$mouse embryonic

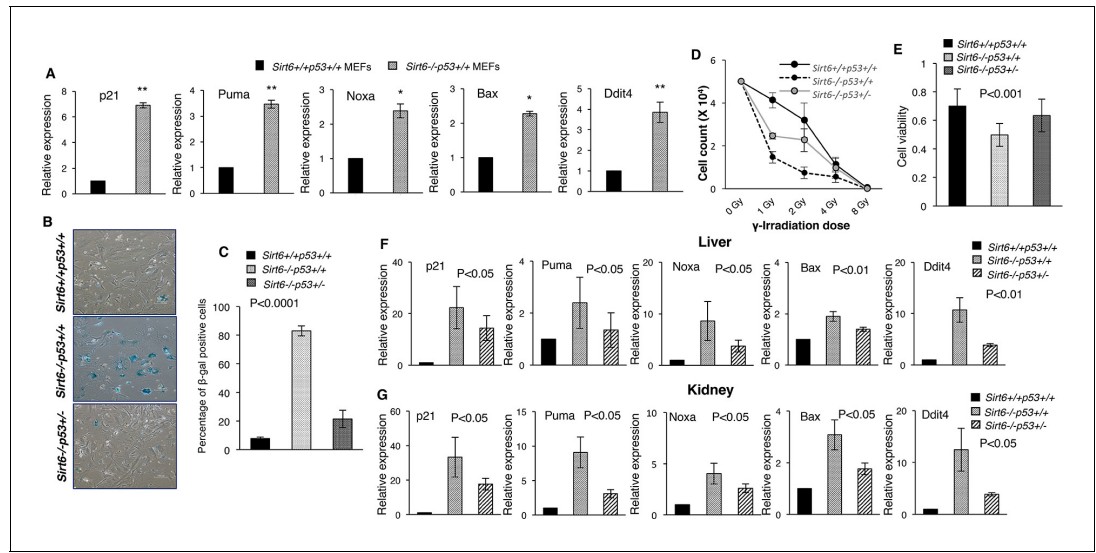

**Figure 1.** Heterozygosity of *Trp53* rescues premature senescence in Sirt6-deficient scenario, (*Trp53* has been denoted as *p53* in the figures). (**A**) Quantification for qPCR analyses for gene expression of p53 targets (with respect to Gapdh controls) in Sirt6$^{+/+}$Trp53$^{+/+}$ (WT) and Sirt6$^{-/-}$Trp53$^{+/+}$ (Sirt6 KO) MEFs, respectively. Data represent mean ± SEM, n = 3. *p<0.05, and **p<0.01 calculated using Student's t-test. (**B**) Representative images of senescence-associated β-galactosidase staining in Sirt6$^{+/+}$Trp53$^{+/+}$, Sirt6$^{-/-}$Trp53$^{+/+}$ and Sirt6$^{-/-}$Trp53$^{+/-}$ MEFs at P6. Scale bar, 100 μm. (**C**) Quantification of data presented in (**B**). Data represent mean ± SEM, approximately 100 cells were counted from each genotype in three replicates. P value calculated using one-way ANOVA. (**D**) Graph showing survival of Sirt6$^{+/+}$Trp53$^{+/+}$, Sirt6$^{-/-}$Trp53$^{+/+}$ and Sirt6$^{-/-}$Trp53$^{+/-}$ MEFs of passage 3 (**P3**) one week after exposure to different doses of γ-irradiation. Data represent mean ± SEM, n=3. (**E**) Graphical representation of cell viability of Sirt6$^{+/+}$Trp53$^{+/+}$, Sirt6$^{-/-}$Trp53$^{+/+}$ and Sirt6$^{-/-}$Trp53$^{+/-}$ MEFs at P3 as measured by MTT assay. Data represent mean ± SEM, n=3. P value calculated using one-way ANOVA. (**F**) Quantification for qPCR analyses of gene expression of p53 targets (with respect to Gapdh controls) in liver from Sirt6$^{+/+}$Trp53$^{+/+}$(WT), Sirt6$^{-/-}$Trp53$^{+/+}$(Sirt6 KO) and Sirt6$^{-/-}$Trp53$^{+/-}$ (compound mutant) mice, respectively. Data represent mean ± SEM, n = 3. P value calculated using one-way ANOVA. (**G**) Quantification for qPCR analyses of gene expression of p53 targets (with respect to Gapdh controls) in kidneys of Sirt6$^{+/+}$Trp53$^{+/+}$(WT), Sirt6$^{-/-}$Trp53$^{+/+}$(Sirt6 KO) and Sirt6$^{-/-}$Trp53$^{+/-}$ (compound mutant) mice, respectively. Data represent mean ± SEM, n = 3. P value calculated using one-way ANOVA.

DOI: https://doi.org/10.7554/eLife.32127.003

The following figure supplement is available for figure 1:

**Figure supplement 1.** Attenuation of p53 downstream targets and rescue of premature senescence-associated phenotypes in Sirt6-deficient scenario by haploinsufficiency of *Trp53*, (*Trp53* has been denoted as *p53* in the figures).

DOI: https://doi.org/10.7554/eLife.32127.004

fibroblasts (MEFs) as compared to Sirt6$^{+/+}$Trp53$^{+/+}$MEFs (*Figure 1A*). Consistent with previous reports stating that loss of Sirt6 results in premature senescence (*Mao et al., 2012*; *Mostoslavsky et al., 2006*), Sirt6$^{-/-}$Trp53$^{+/+}$ MEFs exhibited enhanced senescence-associated β-galactosidase activity and senescence-like flattened cellular morphology at passage 6 (P6) (*Figure 1B and C* and *Figure 1—figure supplement 1A*). Sirt6$^{-/-}$Trp53$^{+/+}$ MEFs also exhibited enhanced p16 levels at P6 (*Figure 1—figure supplement 1B*). However, haploinsufficiency of *Trp53* significantly rescued the senescence-associated phenotypes in Sirt6$^{-/-}$ MEFs at P6 (*Figure 1B and C* and *Figure 1—figure supplement 1A*, 3 independent batches of MEFs have been used in the study which were obtained from independent batches of littermate embryos). Also, *Trp53* haploinsufficiency significantly attenuated p16 levels and the enhanced expression of the downstream targets of p53 in Sirt6$^{-/-}$Trp53$^{+/-}$ MEFs (*Figure 1—figure supplement 1B,C*). Enhancement of sensitivity to DNA damage upon loss of Sirt6 has been previously reported (*Mostoslavsky et al., 2006*). Again, the increased sensitivity to DNA damage in Sirt6$^{-/-}$Trp53$^{+/+}$ MEFs treated with gamma-irradiation was substantially attenuated in Sirt6$^{-/-}$Trp53$^{+/-}$ MEFs (*Figure 1D* and *Figure 1—figure supplement 1D*). The decreased cell viability of Sirt6$^{-/-}$Trp53$^{+/+}$ MEFs was significantly improved in Sirt6$^{-/-}$Trp53$^{+/-}$ MEFs (*Figure 1E*). To further investigate the effects of partial ablation of *p53* in Sirt6 knockout (KO) background at the organismal level, we used compound heterozygous mating strategy to generate Sirt6 KO mice with haploinsufficiency of *p53* (Sirt6$^{-/-}$Trp53$^{+/-}$ mice) as well as Sirt6$^{-/-}$Trp53$^{+/+}$ (Sirt6 KO) and Sirt6$^{+/+}$Trp53$^{+/+}$ (wild-type) littermates. The internal organs, such as kidneys, liver and spleen from these mice were collected for further analyses. Consistent with our findings in MEFs (*Figure 1A*), there was a significant upregulation of the expression of several downstream targets of p53 in the liver, kidneys, and spleen of Sirt6$^{-/-}$Trp53$^{+/+}$ mice (*Figure 1F and G* and *Figure 1E*). However, *Trp53* haploinsufficiency significantly suppressed the expression of those downstream targets of p53 in the liver, kidneys, and spleen of Sirt6$^{-/-}$Trp53$^{+/-}$ mice (*Figure 1F and G* and *Figure 1—figure supplement 1E*). This further suggests that the upregulation of these targets upon loss of Sirt6 is indeed a consequence of p53 activation.

## Heterozygosity of *Trp53* dramatically rescues longevity of Sirt6-deficient mice

Next, we examined whether haploinsufficiency of *Trp53* could rescue the accelerated aging phenotypes of Sirt6-deficient mice. To this end, we employed compound heterozygous mating strategy to mate Sirt6$^{+/-}$Trp53$^{+/+}$ mice (in pure FVB background) with Sirt6$^{+/+}$Trp53$^{+/-}$ mice (in pure C57BL/6 background) to generate Sirt6$^{+/-}$Trp53$^{+/-}$ mice and interbred them to generate Sirt6$^{-/-}$Trp53$^{+/-}$ mice along with Sirt6$^{-/-}$Trp53$^{+/+}$ and Sirt6$^{+/+}$Trp53$^{+/+}$ littermate mice. Given that mixed strains could produce false positive results, we used the litters containing all Sirt6$^{+/+}$Trp53$^{+/+}$, Sirt6$^{-/-}$Trp53$^{+/+}$ and Sirt6$^{-/-}$Trp53$^{+/-}$ mice for further analysis. Consistent with a previous report (*Mostoslavsky et al., 2006*), majority of the Sirt6$^{-/-}$Trp53$^{+/+}$ mice exhibited severe premature aging and were much smaller than their wild-type littermates at 3 weeks of age (*Figure 2A*). Interestingly, the Sirt6$^{-/-}$Trp53$^{+/-}$ mice appeared much healthier than the Sirt6$^{-/-}$Trp53$^{+/+}$ littermates post 3 weeks of birth (*Figure 2A*). Though smaller than their littermate wild-type (Sirt6$^{+/+}$Trp53$^{+/+}$) controls, both female and male Sirt6$^{-/-}$Trp53$^{+/-}$ mice were significantly larger in body size than their Sirt6$^{-/-}$Trp53$^{+/+}$ littermates (*Figure 2A* n=25-35 for all three genotypes). The mean body weights of both female and male Sirt6$^{-/-}$Trp53$^{+/-}$ mice were also significantly more than that of Sirt6$^{-/-}$Trp53$^{+/+}$ littermates (*Figure 2B and C*). The Sirt6-deficient mice mostly died within 4 weeks of birth, whereas the Sirt6$^{-/-}$Trp53$^{+/-}$ littermate mice exhibited striking extensions in lifespan: female mice exhibited a 16-fold increase in maximum lifespan and 11-fold increase in median lifespan (*Figure 2D*); males showed a 14-fold extension in maximum lifespan and a 7.5-fold extension in median lifespan (*Figure 2E*). Given only 4 weeks of lifespan in Sirt6$^{-/-}$Trp53$^{+/+}$mice, the average lifespan of 30 and 44 weeks in male and female Sirt6$^{-/-}$Trp53$^{+/-}$ mice, respectively, represents a significant lifespan extension by *Trp53* haploinsufficiency. Also, the lifespan extension observed in the compound mutant mice in our study (*Figure 2A–E*) was much higher than the lifespan extension observed because of mixed background strains (*Peshti et al., 2017*), thus suggesting that the observed rescue in longevity of sirt6-deficient mice is indeed a result of *Trp53* heterozygosity.

Consistent with previous reports (*Jacks et al., 1994*; *Tyner et al., 2002*), we found that Sirt6$^{+/+}$Trp53$^{+/-}$ mice had median lifespan of 56–64 weeks (*Figure 2D and E*), and mostly succumbed to tumors by 72–80 weeks. However, at four weeks of age, the Sirt6$^{+/+}$Trp53$^{+/-}$ mice exhibited no

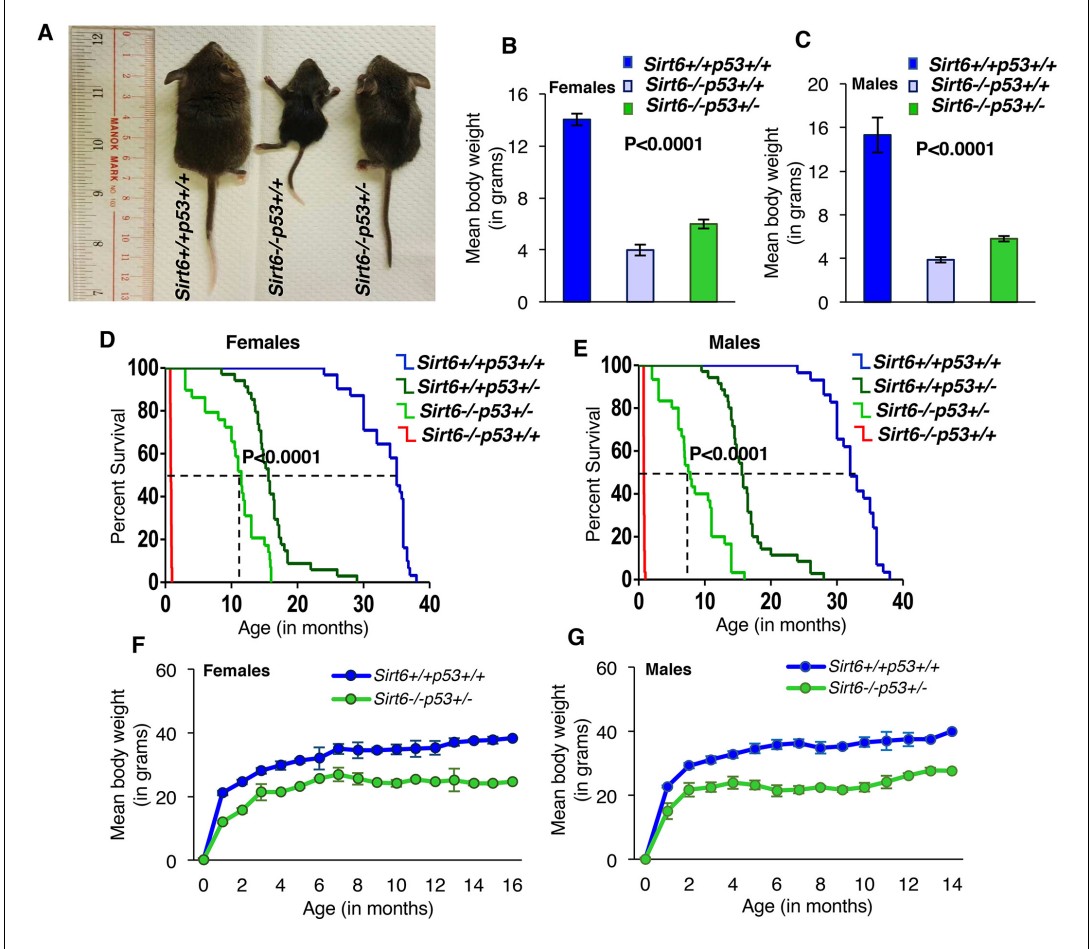

**Figure 2.** Haploinsufficiency of *Trp53* significantly extends the lifespan of Sirt6-deficient mice, (*Trp53* has been denoted as *p53* in the figures). (**A**) Representative images of Sirt6$^{+/+}$Trp53$^{+/+}$(WT), Sirt6$^{-/-}$Trp53$^{+/+}$ (*Sirt6* KO), and $^{-/-}$Trp53$^{+/-}$ (compound mutant) littermate mice at the age of 24 days. (**B**) Mean body weights of WT, *Sirt6* KO and compound mutant female mice at the age of ~24 days. Data represent mean ± SEM; n = 5. P value calculated using one-way ANOVA. (**C**) Mean body weights of WT, *Sirt6* KO and compound mutant male mice at the age of ~24 days. Data represent mean ± SEM; n = 5. P value calculated using one-way ANOVA. (**D**) Kaplan-Meier survival curve of Sirt6$^{+/+}$Trp53$^{+/+}$(n = 30), Sirt6$^{-/-}$Trp53$^{+/+}$ (n = 30), Sirt6$^{-/-}$Trp53$^{+/-}$(n = 31) and Sirt6$^{+/+}$Trp53$^{+/-}$ (n = 35) female mice. P value calculated using log-rank (Mantel-Cox) test. (**E**) Kaplan-Meier survival curve of Sirt6$^{+/+}$Trp53$^{+/+}$ (n = 30), Sirt6$^{-/-}$Trp53$^{+/+}$ (n = 30), Sirt6$^{-/-}$Trp53$^{+/-}$ (n = 31) and Sirt6$^{+/+}$Trp53$^{+/-}$ (n = 35) male mice. P value calculated using log-rank (Mantel-Cox) test. (**F**) Graphical representation of increasing body weights in female Sirt6$^{-/-}$Trp53$^{+/-}$ and WT littermate mice. Data represent mean ± SEM. (**G**) Graphical representation of increasing body weights in male Sirt6$^{-/-}$Trp53$^{+/-}$ and WT littermate mice. Data represent mean ± SEM.

DOI: https://doi.org/10.7554/eLife.32127.005

The following source data and figure supplement are available for figure 2:

**Source data 1.** Serum Igf-1 levels in 24-day old mice.
DOI: https://doi.org/10.7554/eLife.32127.007

**Source data 2.** Serum Igf-1 levels in 10-16 months old mice.
DOI: https://doi.org/10.7554/eLife.32127.008

**Source data 3.** Serum glucose levels in 24-day old mice.
DOI: https://doi.org/10.7554/eLife.32127.009

**Source data 4.** Serum glucose levels in 10-16 months old mice.
DOI: https://doi.org/10.7554/eLife.32127.010

**Figure supplement 1.** Analysis of serum IGF-1 and glucose levels in Sirt6$^{+/+}$Trp53$^{+/+}$, Sirt6$^{-/-}$Trp53$^{+/+}$ and Sirt6$^{-/-}$Trp53$^{+/-}$ mice, (Trp53 has been denoted as *p53* in the figures).
DOI: https://doi.org/10.7554/eLife.32127.006

significant differences from Sirt6$^{+/+}$Trp53$^{+/+}$ (wild-type) mice in either body size or body weight. The body weights of Sirt6$^{-/-}$Trp53$^{+/-}$ mice (both females and males) increased steadily with age as did their wild-type (WT) littermates, although they always weighed less than their WT littermate mice (*Figure 2F and G*). At 3 weeks of age, the Sirt6$^{-/-}$Trp53$^{+/-}$ mice were more active than the Sirt6$^{-/-}$Trp53$^{+/+}$mice but were less active than their WT littermates. While comparing p53 expression levels in the cells and tissues of 3-week old mice, p53 protein level in Sirt6$^{-/-}$Trp53$^{+/-}$ mice was observed to be decreased as compared to Sirt6$^{-/-}$Trp53$^{+/+}$ mice, but was more than that of Sirt6$^{+/+}$Trp53$^{+/+}$ mice (*Figure 1—figure supplement 1F*).

Next, we analyzed the GH/IGF1 axis in the mice by checking their serum IGF1 levels. As previously reported (*Mostoslavsky et al., 2006*), the serum IGF1 levels were drastically reduced in Sirt6$^{-/-}$Trp53$^{+/+}$ mice as compared to Sirt6$^{+/+}$Trp53$^{+/+}$ mice at the age of around 24 days . Interestingly, the serum IGF1 levels in Sirt6$^{-/-}$Trp53$^{+/-}$ mice were significantly rescued with respect to Sirt6$^{-/-}$Trp53$^{+/+}$ mice, but were less than that of their wild-type littermates (*Figure 2—figure supplement 1A*). Also, the serum IGF1 levels of Sirt6$^{-/-}$Trp53$^{+/-}$ mice were less than that of their wild-type littermates during their terminal stage (10-16 months) (*Figure 2—figure supplement 1B*). Apart from serum IGF1 levels, we also analyzed glucose levels in the mice. The Sirt6$^{-/-}$Trp53$^{+/+}$ mice displayed a minor drop in their serum glucose levels, while the compound mutant (Sirt6$^{-/-}$Trp53$^{+/-}$) mice exhibited an almost comparable glucose level to their wild-type littermates at the age of 24 days (*Figure 2—figure supplement 1C*). Although Sirt6$^{-/-}$Trp53$^{+/-}$ mice, in their terminal lifespan, had slightly reduced serum glucose levels than their wild-type littermates, the values did not reach significance (*Figure 2—figure supplement 1D*).

## Amelioration of aging-associated phenotypes in Sirt6-deficient mice upon haploinsufficiency of *Trp53*

The dramatic extension of lifespan in Sirt6-deficient mice upon haploinsufficiency of *Trp53* encouraged us to further examine whether the premature aging-associated abnormalities previously reported in Sirt6-deficient mice (*Mostoslavsky et al., 2006*) are rescued or not. As expected, at around 24 days of age, the sizes of the internal organs of Sirt6$^{-/-}$Trp53$^{+/-}$ mice, such as spleen, thymus, kidney, liver, lungs and heart, were markedly larger than those of Sirt6$^{-/-}$Trp53$^{+/+}$ mice, although they were smaller than those of wild-type (Sirt6$^{+/+}$Trp53$^{+/+}$) littermates (*Figure 3A*). Lordokyphosis (increased curvature of the spine), a prominent phenotype of Sirt6$^{-/-}$Trp53$^{+/+}$ mice (*Mostoslavsky et al., 2006*), was significantly improved in Sirt6$^{-/-}$Trp53$^{+/-}$mice, irrespective of gender (*Figure 3B* and *Figure 3—figure supplement 1A*). Colitis, a characteristic feature previously reported in Sirt6-deficient mice (*Mostoslavsky et al., 2006*), was largely rescued in Sirt6$^{-/-}$Trp53$^{+/-}$ mice when compared to the Sirt6$^{-/-}$Trp53$^{+/+}$ littermates (*Figure 3C*). When senescence was evaluated, there was significantly less senescence-associated β-galactosidase activity in the organs from Sirt6$^{-/-}$Trp53$^{+/-}$ mice, such as spleen, liver and kidneys, as compared to Sirt6$^{-/-}$Trp53$^{+/+}$ mice (*Figure 3D*), thus suggesting a marked rescue of premature senescence in the compound mutant mice upon haploinsufficiency of *Trp53*. Taken together, these results are in line with our observation of lifespan extension in *Sirt6*-deficient mice by haploinsufficiency of *Trp53*, and further confirm that elevated p53 activity plays a significant role in accelerated aging and premature death in *Sirt6*-deficient mice.

Regarding cancer incidence, some of the Sirt6$^{+/+}$Trp53$^{+/+}$ (WT) mice (approximately 6–8%) developed tumors at around 2–2.5 years, as expected. Approximately 26–28% of Sirt6$^{+/+}$Trp53$^{+/-}$ mice developed tumors by 17–20 months, regardless of gender as already reported (*Jacks et al., 1994*; *Tyner et al., 2002*). The Sirt6$^{-/-}$Trp53$^{+/+}$ mice mostly died within 4 weeks of birth, and no incidence of cancer was observed. Interestingly, around 32% of the Sirt6$^{-/-}$Trp53$^{+/-}$ (compound mutant) mice developed tumors during their terminal lifespan. However, the incidence of cancer in Sirt6$^{-/-}$Trp53$^{+/-}$ mice was earlier (9-16 months) than the Sirt6$^{+/+}$Trp53$^{+/-}$ mice (17-20 months). The Sirt6$^{+/+}$Trp53$^{+/-}$ mice mostly developed sarcoma by 17-20 months (*Figure 3—figure supplement 1B*). However, the Sirt6$^{-/-}$Trp53$^{+/-}$ mice developed other types of tumors, such as kidney tumor (19%), prostate tumor (17%), bladder tumor (16%), and pancreatic tumor (8%) (*Figure 3—figure supplement 1C–F* respectively), while only 3% of them exhibited incidence of sarcoma. Taken together, sirt6 depletion enhanced the onset and diversified the type of tumors in the Trp53$^{+/-}$ background, given that Sirt6 is a tumor suppressor itself (*Lerrer and Cohen, 2013*; *Sebastián et al., 2012* ).

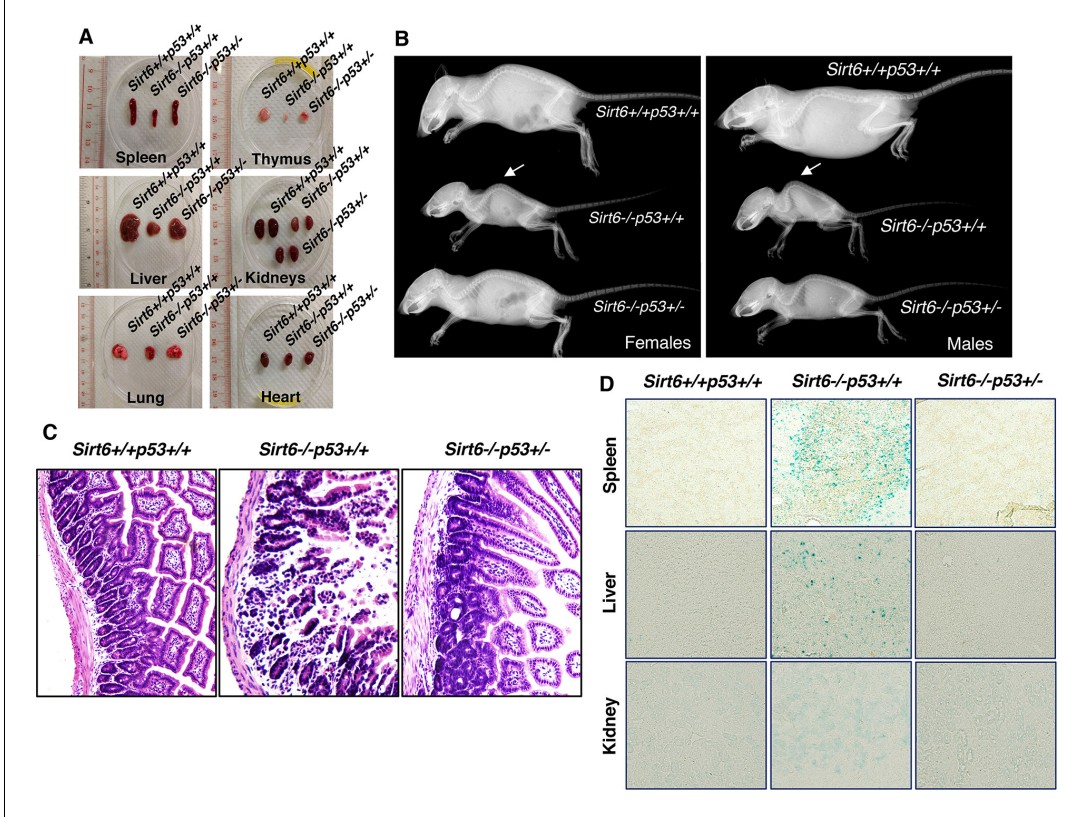

**Figure 3.** Amelioration of premature-aging associated phenotypes in *Sirt6* KO mice upon haploinsufficiency of *p53*, (*Trp53* has been denoted as *p53* in the figures). (A) Representative images of spleens, thymus, livers, kidneys, lungs and hearts from 24 days old Sirt6$^{+/+}$Trp53$^{+/+}$, Sirt6$^{-/-}$Trp53$^{+/+}$, and Sirt6$^{-/-}$Trp53$^{+/-}$ mice. (B) Whole body X-ray images of female and male Sirt6$^{+/+}$Trp53$^{+/+}$, Sirt6$^{-/-}$Trp53$^{+/+}$ and Sirt6$^{-/-}$Trp53$^{+/-}$ mice (note the lordokyphosis/curved spine in Sirt6$^{-/-}$p53$^{+/+}$ mice) at the age of 24 days. (C) Histology of small intestinal cross-sections of WT, *Sirt6* KO and compound mutant mice to detect colitis, at the age of ~24 days. (D) Representative images of senescence-associated β-galactosidase staining in spleen, liver and kidneys of 3 weeks old Sirt6$^{+/+}$Trp53$^{+/+}$ (WT), Sirt6$^{-/-}$Trp53$^{+/+}$ (Sirt6 KO) and Sirt6$^{-/-}$Trp53$^{+/-}$ (compound mutant) mice.

DOI: https://doi.org/10.7554/eLife.32127.011

The following figure supplement is available for figure 3:

**Figure supplement 1.** Incidence of cancer and other age-associated disorders in Sirt6$^{-/-}$Trp53$^{+/-}$ mice, (*Trp53* has been denoted as *p53* in the figures).
DOI: https://doi.org/10.7554/eLife.32127.012

Apart from severe body wasting and loss of vigour in Sirt6$^{-/-}$Trp53$^{+/-}$ mice in their terminal stage, some female Sirt6$^{-/-}$Trp53$^{+/-}$ mice succumbed to the disorder of incontinence at around 14-16 months, while around 95% of the Sirt6$^{-/-}$Trp53$^{+/-}$ male mice exhibited penile protrusion at around 9-11 months of age (*Figure 3—figure supplement 1G*). Also, the Sirt6$^{-/-}$Trp53$^{+/-}$ mice remained sterile throughout their lifespan.

## Rescue of bone marrow-derived stem cell decline and deteriorated immune cell count in Sirt6-deficient mice upon heterozygosity of *Trp53*

Stem cell depletion is one of the hallmarks of premature aging and heterozygosity of *Trp53* has been reported to improve stem cell maintenance (*Amir et al., 2017*; *Belle et al., 2015*; *López-Otín et al., 2013*; *Lu et al., 2015*).We therefore sought to examine the status of mesenchymal and hematopoietic stem cells in the bone marrow of Sirt6$^{+/+}$Trp53$^{+/+}$, Sirt6$^{-/-}$Trp53$^{+/+}$and Sirt6$^{-/-}$Trp53$^{+/-}$ mice (isolation and purification of bone marrow-derived stem cells have been described in the methods). Consistent with the age-associated decline of bone marrow stem cells, we observed a notable drop in the number of mesenchymal and hematopoietic stem cells derived from the bone marrow of both male and female Sirt6$^{-/-}$Trp53$^{+/+}$ mice (*Figure 4A–D*). However, upon *Trp53* haploinsufficiency, the percentage of mesenchymal and hematopoietic stem cells in the bone marrow of Sirt6$^{-/-}$Trp53$^{+/-}$

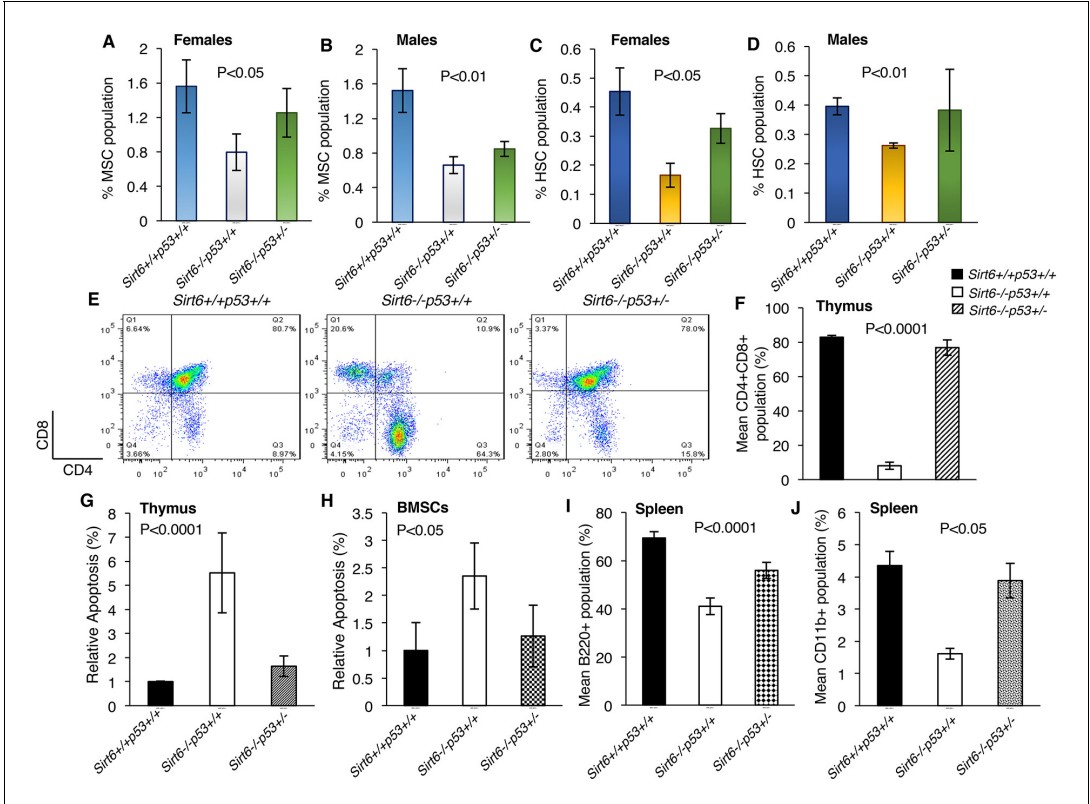

**Figure 4.** Haploinsufficiency of *Trp53* prevents bone marrow stem cell decline, suppresses apoptosis, and restores immune cell populations in Sirt6-deficient mice, (*Trp53* has been denoted as *p53* in the figures). (**A**) Percentage of mesenchymal stem cell (MSC) population in the female Sirt6$^{+/+}$Trp53$^{+/+}$, Sirt6$^{-/-}$Trp53$^{+/+}$ and Sirt6$^{-/-}$Trp53$^{+/-}$ mice at the age of ~24 days. Data represent mean ± SEM; n=5. (**B**) Percentage of mesenchymal stem cell (MSC) population in the male Sirt6$^{+/+}$Trp53$^{+/+}$, Sirt6$^{-/-}$Trp53$^{+/+}$ and Sirt6$^{-/-}$Trp53$^{+/-}$ mice at the age of ~24 days. Data represent mean ± SEM; n=5. (**C**) Percentage of hematopoietic stem cell (HSC) population in the female Sirt6$^{+/+}$Trp53$^{+/+}$, Sirt6$^{-/-}$Trp53$^{+/+}$ and Sirt6$^{-/-}$Trp53$^{+/-}$ mice at the age of ~24 days. Data represent mean ± SEM; n=5. (**D**) Percentage of hematopoietic stem cell (HSC) population in the male Sirt6$^{+/+}$Trp53$^{+/+}$, Sirt6$^{-/-}$Trp53$^{+/+}$ and Sirt6$^{-/-}$Trp53$^{+/-}$ mice at the age of ~24 days. Data represent mean ± SEM; n=5. (**E**) Representative flow cytometry analysis of CD4$^+$CD8$^+$ cells in the thymus of Sirt6$^{+/+}$Trp53$^{+/+}$, Sirt6$^{-/-}$Trp53$^{+/+}$ and Sirt6$^{-/-}$Trp53$^{+/-}$ mice at the age of ~24 days. Y-axis and X-axis represent the numbers of CD8$^+$ cells and CD4$^+$ cells respectively. (**F**) Graphical representation of data presented in (**E**). Data represent mean ± SEM; n = 5. (**G**) Quantification of apoptotic cells in the thymus of Sirt6$^{+/+}$Trp53$^{+/+}$, Sirt6$^{-/-}$Trp53$^{+/+}$ and Sirt6$^{-/-}$Trp53$^{+/-}$ mice by Annexin V staining. Data represent mean ± SEM; n=4. (**H**) Quantification of apoptotic cells in the thymus and bone marrow stroma of Sirt6$^{+/+}$Trp53$^{+/+}$, Sirt6$^{-/-}$Trp53$^{+/+}$ and Sirt6$^{-/-}$Trp53$^{+/-}$ mice by Annexin V staining. Data represent mean ± SEM; n=4. (**I**) Graphical representation of peripheral B cell population (B220$^+$ cells) in the spleens from Sirt6$^{+/+}$Trp53$^{+/+}$, Sirt6$^{-/-}$Trp53$^{+/+}$ and Sirt6$^{-/-}$Trp53$^{+/-}$ mice. Data represent mean ± SEM; n=5. (**J**) Graphical representation of peripheral monocyte counts (CD11b$^+$ cells) in the spleens from Sirt6$^{+/+}$Trp53$^{+/+}$, Sirt6$^{-/-}$Trp53$^{+/+}$ and Sirt6$^{-/-}$Trp53$^{+/-}$ mice. Data represent mean ± SEM; n=5. All P values have been calculated using one-way ANOVA.

DOI: https://doi.org/10.7554/eLife.32127.013

mice (both females and males) was partially, yet significantly restored as compared to their littermate Sirt6$^{-/-}$Trp53$^{+/+}$ mice (*Figure 4A–D*).

It has been previously reported that CD4$^+$CD8$^+$ double-positive cells undergo drastic reduction in the thymus of Sirt6$^{-/-}$Trp53$^{+/+}$mice post 3 weeks of birth (*Mostoslavsky et al., 2006*). Consistently, we also observed a marked decline in the thymic CD4$^+$CD8$^+$ double-positive cells of Sirt6$^{-/-}$Trp53$^{+/+}$ mice at around 24 days of age (*Figure 4E and F*). However, there was a striking increase in the number of CD4$^+$CD8$^+$ double-positive cells in the thymus of Sirt6$^{-/-}$Trp53$^{+/-}$ mice as compared to that in the Sirt6$^{-/-}$Trp53$^{+/+}$ littermates (*Figure 4E and F*). We confirmed the previous observation that apoptotic responses are upregulated in the thymus of Sirt6-deficient mice (*Mostoslavsky et al., 2006*) (*Figure 4G*). Interestingly, this was partially rescued in both thymus and bone marrow stromal cells (BMSCs) of Sirt6$^{-/-}$Trp53$^{+/-}$ mice when compared to Sirt6$^{-/-}$Trp53$^{+/+}$ mice (*Figure 4G and H*). In addition, there was a significant increase in the number of B220$^+$ cells (characterizing B cells across all developmental stages) and CD11b$^+$ cells (representing monocytes) in the spleen of Sirt6$^{-/-}$Trp53$^{+/-}$

mice than that in Sirt6$^{-/-}$Trp53$^{+/+}$ littermates (*Figure 4I and J*), thus indicating that haploinsufficiency of *Trp53* rescues the total B cell count and the total monocyte count, respectively, in Sirt6-deficient scenario.

## SIRT6 is a NAD$^+$-dependent deacetylase for p53 at lysine 381, but not lysine 382

To further investigate the mechanistic link between Sirt6 deficiency and p53 activation and understand how p53 is involved in the accelerated aging of Sirt6-deficient mice, we first tested if these two proteins physically interact. To this end, we performed reciprocal co-immunoprecipitation experiments to evaluate interactions between SIRT6 and p53 in HEK293 cells without any external DNA damage, since DNA damage-mediated p53 upregulation could taint the results on interaction. Via co-immunoprecipitation with specific antibodies against SIRT6 and p53, we observed interaction between the two proteins at the endogenous level (*Figure 5A*). To further confirm this interaction, we ectopically expressed FLAG-tagged SIRT6 and p53 in HEK293 cells individually, and could

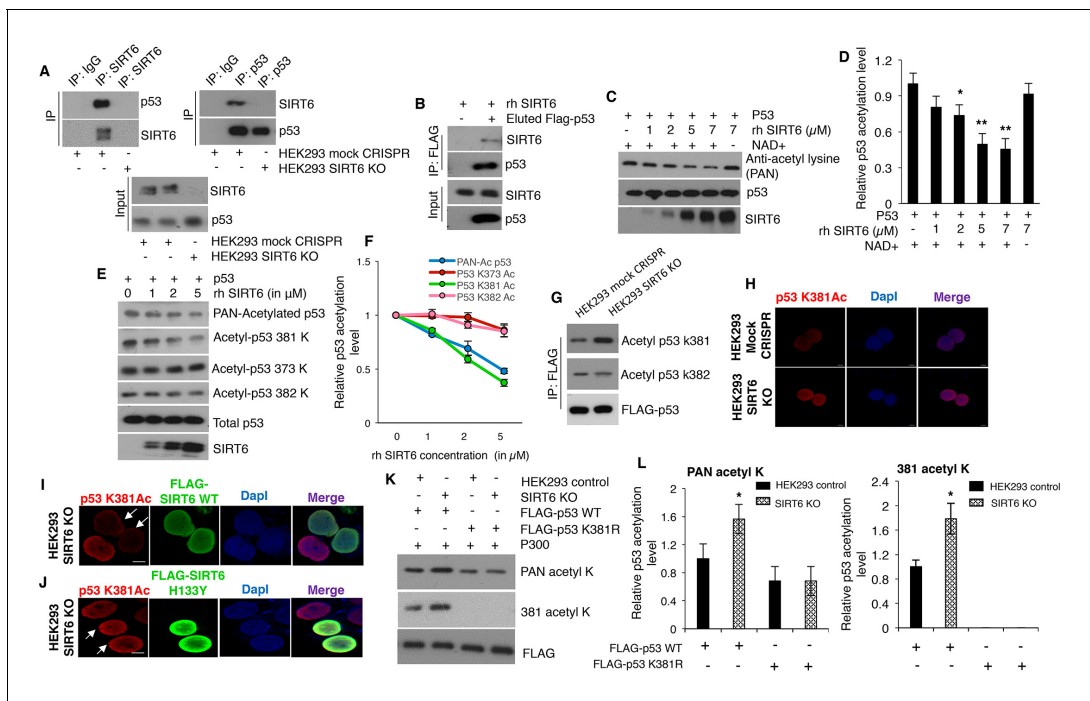

**Figure 5.** SIRT6 directly deacetylates p53 at lysine 381, but not K382. (**A**) Western blotting analysis of endogenous interaction between SIRT6 and p53 in mock CRISPR control cells and SIRT6 KO cells using specific antibodies for SIRT6 and p53, keeping respective IgG controls. (**B**) Western blotting analysis to examine direct interaction between SIRT6 and p53 in vitro using recombinant human (rh) SIRT6 and eluted FLAG-tagged p53 from HEK293 cells. (**C**) Western blotting analysis of deacetylation of p53 in vitro with increasing concentration of recombinant (rh) SIRT6 and purified p53, in the presence and absence of NAD$^+$. (**D**) Quantification of data presented in (**C**). Data represent mean ± SEM, n = 3. calculated using Student's t-test. (**E**) Analyses of SIRT6-mediated deacetylation of p53 at lysine (**K**) 381, 373 or 382 in vitro by Western blotting using specific antibodies and PAN-acetyl lysine antibodies. (**F**) Quantification of data presented in (**E**). Data represent mean ± SEM, n = 3. (**G**) Western blotting data showing acetylation at lysine 381 and 382 in immunoprecipitated FLAG-tagged p53 in SIRT6 KO cells and mock CRISPR control HEK293 cells. (**H**) Immunofluorescence staining of p53 acetylation at K381 in HEK293 mock CRISPR and SIRT6 KO cells. Scale bar, 10 μm. (**I**) Immunofluorescence staining of p53 acetylation at K381 (red fluorescence) in SIRT6 KO cells with ectopically expressed wild-type (WT) SIRT6 (green fluorescence). Scale bar, 10 μm. (**J**) Immunofluorescence staining of p53 acetylation at K381 (red fluorescence) in SIRT6 KO cells with ectopically expressed catalytically inactive mutant SIRT6 (H133Y) (green fluorescence). Scale bar, 10 μm. (**K**) Western blotting analysis of SIRT6-mediated deacetylation of WT and lysine 381 to arginine (K381R) mutant p53 by antibodies against pan-acetyl lysine and acetylation at lysine 381 of p53. (**L**) Quantification of data presented in (**K**). Data represent mean ± SEM, n = 3. *p<0.05 calculated using Student's t-test.

DOI: https://doi.org/10.7554/eLife.32127.014

The following figure supplement is available for figure 5:

**Figure supplement 1.** Analyses of physical and functional interaction between SIRT6 and p53, (*Trp53* has been denoted as *p53* in the figures).

DOI: https://doi.org/10.7554/eLife.32127.015

observe endogenous p53 and SIRT6 being pulled down in FLAG immunoprecipitates respectively (*Figure 5—figure supplement 1A and B*). Next, to rule out the possibility of cell-specific effect on the interaction between SIRT6 and p53, we used recombinant human SIRT6 (rh SIRT6) and FLAG-p53 eluted from HEK293 cells for in vitro binding experiment and analyzed direct physical association between SIRT6 and p53. Indeed, p53 could pull down SIRT6 in vitro, suggesting a direct interaction between SIRT6 and p53 (*Figure 5B*).

Acetylation is one of the major post-translational modifications regulating the stability and activity of p53 (*Marouco et al., 2013*; *Reed and Quelle, 2014*). A previous study has indicated that SIRT6, as a NAD$^+$-dependent deacetylase, is not responsible for p53 deacetylation at lysine (K) 382 (*Michishita et al., 2005*). However, given that multiple potential acetylation sites exist in p53 protein (*Marouco et al., 2013*; *Reed and Quelle, 2014*), it is conceivable to speculate that SIRT6 may deacetylate p53 at sites other than K382. To test this, we performed in vitro deacetylation assays using eluted FLAG-p53 (purified from HEK293 cells) and recombinant SIRT6 (rh SIRT6). As predicted, we observed that SIRT6 could deacetylate p53 in vitro (*Figure 5C*). In line with the nature of SIRT6 as a NAD$^+$-dependent deacetylase, SIRT6-mediated p53 deacetylation in vitro was abrogated in the absence of NAD$^+$ (*Figure 5C and D*). In addition, this deacetylation was largely diminished in the presence of the sirtuin inhibitor nicotinamide (*Figure 5—figure supplement 1C and D*). In accordance with existing literature (*Michishita et al., 2005*), we observed no alteration in p53 acetylation at lysine (K) 382 even with increasing concentration of rhSIRT6 in vitro (*Figure 5E*). However, when checking the other two major sites of acetylation in the C-terminus of p53, that is lysine 381 and 373 (*Gu and Roeder, 1997*), the pan-acetylation (*i.e.*, total acetylation) and acetylation at K381 of p53 was found to significantly reduce with increasing concentrations of SIRT6 (*Figure 5E and F*). No significant change in the acetylation of p53 at K373 was observed in the presence of rhSIRT6 (*Figure 5E and F*). Consistently, p53 acetylation at K381 was upregulated in SIRT6-deficient cells (SIRT6 KO cells) in comparison with mock CRISPR control cells, when assessed with western blotting (*Figure 5G*). Again, no significant change in p53 acetylation at K382 was observed in SIRT6-deficient cells when compared with mock CRISPR control cells (*Figure 5G*). In agreement with Western blotting results, immunofluorescence staining also showed that p53 acetylation was upregulated at K381, but not K382 in SIRT6-deficient cells (*Figure 5H* and *Figure 5—figure supplement 1E*). When p53 lysine 381 was mutated to arginine (K381R, non-acetylatable mutant), p53 acetylation at this site was no longer detectable, thus confirming the epitope specificity of the antibodies against p53 acetylation at K381 (*Figure 5—figure supplement 1F*). Additionally, expression of the FLAG-tagged wild-type (WT) form of SIRT6 attenuated the upregulated p53 K381 acetylation in SIRT6-deficient HEK293 cells (*Figure 5I*). However, such attenuation in p53 acetylation at K381 was not observed with the ectopic expression of the catalytically inactive form of SIRT6 (H133Y) (*Figure 5J*). Although 90–95% of the SIRT6 KO cells were stained with antibodies against p53 acetylated at K382, ectopically expressed SIRT6 (both active and catalytically inactive forms) had no effect on p53 acetylation at K382 (*Figure 5—figure supplement 1G*). Next, we analyzed immunoprecipitated FLAG-tagged WT p53 and K381R mutant p53 for acetylation in mock CRISPR control cells and SIRT6-deficient cells. Consistent with data shown in *Figure 5G*, pan-acetylation of WT p53 was increased in the SIRT6-deficient cells over control cells (*Figure 5K and L*) and the K381R mutation caused a significant reduction in pan-acetylation of p53 in both control and SIRT6-deficient cells (*Figure 5K*). Surprisingly, the increase in pan-acetylation of p53 was largely diminished for K381R mutant in SIRT6-deficient cells as compared to control cells (*Figure 5K and L*). These findings support the idea that the increased p53 pan-acetylation in the absence of SIRT6 is largely attributable to the acetylation status at lysine 381 of p53 (*Figure 5K and L*). Also, endogenous p53 immunoprecipitated from SIRT6 KO HEK293 cells exhibited more acetylation at K381 as compared to mock CRISPR cells, and this increased acetylation of p53 at K381 went down with ectopic expression of wild-type (WT) SIRT6 (*Figure 5—figure supplement 1H*). To further substantiate the observation of upregulation of p53 acetylation at K381 upon Sirt6 deficiency, we tested the same in cells and tissues of mice, and similarly observed upregulation of p53 acetylation at K381 in Sirt6$^{-/-}$Trp53$^{+/+}$ mice as compared to Sirt6$^{+/+}$Trp53$^{+/+}$ littermate mice (*Figure 5—figure supplement 1I and J*).

## SIRT6 negatively regulates the stability of p53

It is known that acetylation enhances the stability of p53 (*Gu and Zhu, 2012*; *Kruse and Gu, 2009*). Indeed, HEK293 cells lacking SIRT6 (SIRT6 KO cells) had approximately two-fold increases in the

total p53 protein level when compared with mock CRISPR control cells (*Figure 6A and B*). Immuno-fluorescence staining further confirmed the increased endogenous levels of p53 in SIRT6 KO cells as compared to control cells (*Figure 6C*). Similarly, liver and kidney tissues from Sirt6-deficient mice exhibited a significant increase in p53 protein levels (*Figure 6D and E*). This increase in p53 expression was also observed in Sirt6 null (Sirt6$^{-/-}$) primary mouse embryonic fibroblasts (passage P2 MEFs were used to avoid alterations because of serial passaging) (Figures *Figure 6—figure supplement 1A and B*). Treatment with the protein synthesis inhibitor cycloheximide further confirmed the enhanced stability of p53 in SIRT6-deficient cells (Figures *Figure 6—figure supplement 1C and D*). However, we did not observe increased phosphorylation of p53 at serine 15 in SIRT6-deficient

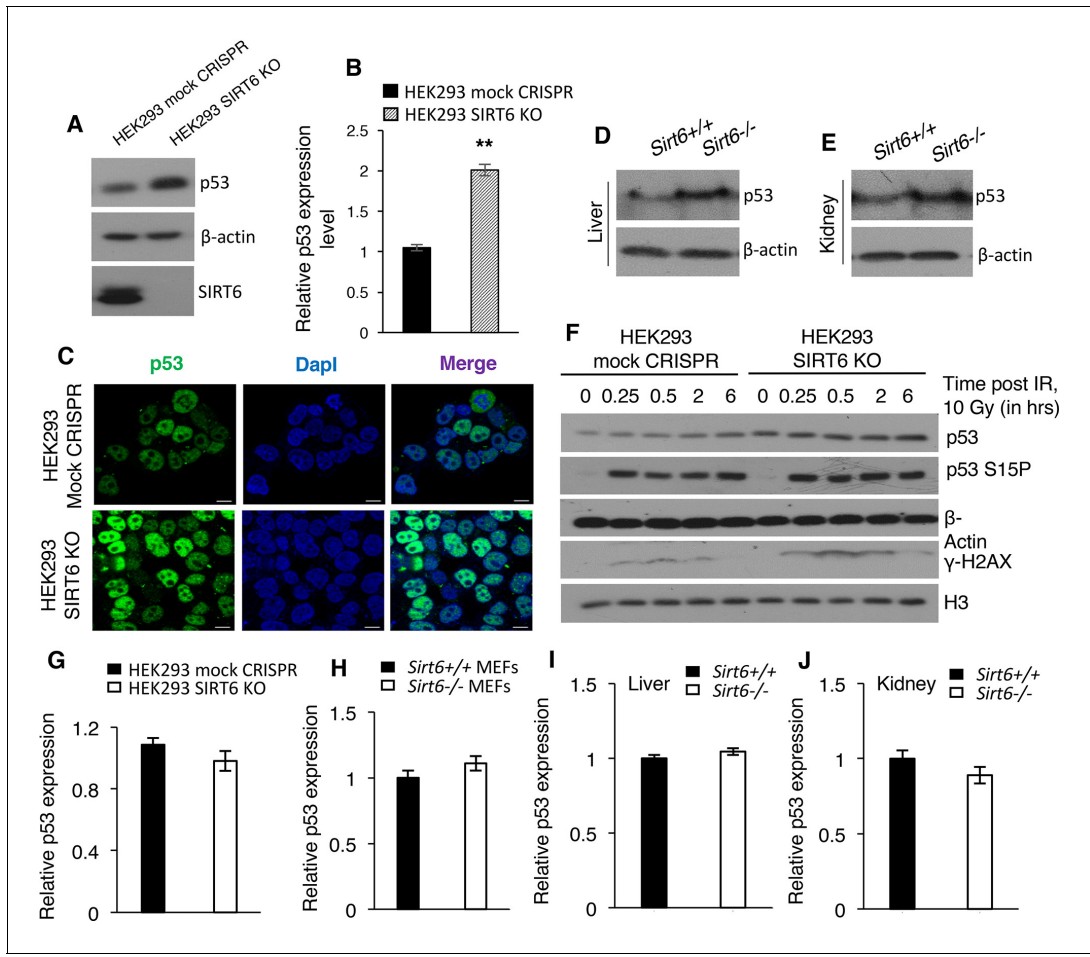

**Figure 6.** SIRT6 negatively regulates the stability of p53. (A) Western blotting analysis of p53 protein expression in SIRT6 KO and control HEK293 cells. (B) Quantification of data presented in (A). Data represent mean ± SEM, n = 3. **p<0.01 calculated using Student's t-test. (C) Immunofluorescence staining to confirm enhanced p53 expression in HEK293 mock CRISPR and SIRT6 KO cells. Scale bar, 10 μm. (D) Western blotting analysis of p53 protein expression in liver of wild-type (WT) and Sirt6$^{-/-}$ mice. (E) Western blotting analysis of p53 protein expression in kidneys of wild-type (WT) and Sirt6$^{-/-}$ mice. (F) Western blotting analysis of p53 and phosphorylation of p53 at serine 15 in HEK293 mock CRISPR and SIRT6 KO cells in response to 10 Gy of γ-irradiation. (G) qPCR analysis of *p53* expression in HEK293 mock CRISPR and SIRT6 KO cells (with respect to Gapdh controls). Data represent mean ± SEM, n = 3. (H) qPCR analysis of *p53* expression in Sirt6$^{+/+}$ and Sirt6$^{-/-}$ MEFs (with respect to Gapdh controls). Data represent mean ± SEM, n = 3. (I) qPCR analysis of *p53* expression in liver of Sirt6$^{+/+}$ and Sirt6$^{-/-}$ mice (with respect to Gapdh controls). Data represent mean ± SEM, n = 3. (J) qPCR analysis of *p53* expression in kidneys of Sirt6$^{+/+}$ and Sirt6$^{-/-}$ mice (with respect to Gapdh controls). Data represent mean ± SEM, n = 3.

DOI: https://doi.org/10.7554/eLife.32127.016

The following figure supplements are available for figure 6:

**Figure supplement 1.** Analyses of the stability of p53 in the presence and absence of SIRT6.
DOI: https://doi.org/10.7554/eLife.32127.017

**Figure supplement 2.** Acetyl mimic mutant p53 K381Q confers stability to p53 and imparts senescence-like properties to cells.
DOI: https://doi.org/10.7554/eLife.32127.018

HEK293 cells compared with wild-type cells without irradiation (*Figure 6F*). This phosphorylation of p53 robustly increased in both wild type and Sirt6-deficient cells upon irradiation (*Figure 6F*). This suggests that the upregulation of p53 in SIRT6-deficient cells is unlikely a direct consequence of increased DNA damage in the absence of SIRT6 (*Figure 6F*). Furthermore, we observed no significant increase in p53 mRNA levels in the absence of Sirt6 in HEK293 cells, MEF cells or mice tissues (*Figure 6G–J*). Hence, it is likely that SIRT6 regulates p53 at the post-translational level.

Since we observed that SIRT6 deacetylates p53 at lysine 381 and loss of SIRT6 conferred stability to p53, we further sought to examine the importance of p53 acetylation at K381 on the stability of p53. To address this, we generated another mutant construct of p53, with lysine 381 mutated to glutamine (K381Q) by site-directed mutagenesis, which mimics the acetyl form of p53 at K381. FLAG-tagged WT p53, K381R mutant (lysine to arginine, non-acetylatable mutant), and K381Q mutant (lysine to glutamine, acetyl mimic mutant) were ectopically expressed in HEK293 cells individually, followed by treatment with cycloheximide to examine the stability of different mutants of p53. As predicted, we observed a significant drop in the stability of p53 K381R mutant, which was further intensified upon cycloheximide treatment (*Figure 6—figure supplement 2A and B*). On the other hand, p53 K381Q mutant displayed improved stability, as evidenced by the comparable protein expression levels before and after cycloheximide treatment (*Figure 6—figure supplement 2A and B*). These findings reinstate that loss of SIRT6 leads to upregulation of p53 acetylation at K381, which further confers stability to p53. In agreement with increase in the expression of senescent biomarker p16 in SIRT6 KO cells with respect to control cells, we also observed an increase in the expression of p16 in cells with ectopic expression of p53 K381Q mutant (*Figure 6—figure supplement 2C and D*). This data suggests that hyperacetylation of p53 at K381 imparts senescence-like properties to cells, thus contributing to the premature senescence observed in Sirt6-deficient cells.

## Discussion

Although recent studies have reported that p53 upregulates SIRT6 expression (*Jung et al., 2016*; *Zhang et al., 2014*), the mechanistic explanation of this regulation remains largely unclear. On the other hand, there is no existing literature about the regulation of p53 by SIRT6. In this study, we show that loss of SIRT6 results in increased acetylation of p53, leading to elevated apoptosis and senescence at the cellular level and to accelerated aging in mice. Haploinsufficiency of *Trp53* significantly reduced premature senescence, substantially restored immune cell and stem cell populations, and dramatically extended the lifespan of Sirt6-deficient mice. These data identified a novel SIRT6-p53 axis in the regulation of senescence and aging. Our study reveals a direct link between SIRT6 and p53, wherein the stability of p53 is regulated through SIRT6-mediated deacetylation of p53 at K381. Hence, these findings not only establish p53 as a substrate for SIRT6, but also identify SIRT6-mediated p53 deacetylation as a critical mechanism to suppress cellular senescence and organismal aging. Although we have identified lysine 381 of p53 as a major site of deacetylation by SIRT6, we cannot exclude the possibility that SIRT6 might target other lysine residues of p53 for deacetylation or other post-translational modifications. This would require further investigation in the future.

Sirt6$^{-/-}$Trp53$^{+/+}$ mice have been reported to die because of acute aging-associated degenerative processes, such as severe organ degeneration, body wasting, abruptly lowered serum IGF-1 levels, gut inflammation and chronic metabolic disorders (*Mostoslavsky et al., 2006*). The Sirt6$^{-/-}$Trp53$^{+/-}$ mice showed signs of severe body wasting during their terminal lifespan (around 12-16 months). Around 32% of the Sirt6$^{-/-}$Trp53$^{+/-}$ (compound mutant) mice also developed cancer during their terminal lifespan. Some Sirt6$^{-/-}$Trp53$^{+/-}$ female mice succumbed to the disorder of incontinence, that is uncontrolled urination at around 14-16 months, while around 95% of Sirt6$^{-/-}$Trp53$^{+/-}$ male mice exhibited penile protrusion at around 9-11 months of age (*Figure 1G*). Taken together, the Sirt6$^{-/-}$Trp53$^{+/-}$ mice died of disorders which are both overlapping with and distinct from Sirt6$^{-/-}$Trp53$^{+/+}$ mice.

Although mixed genetic background has been reported to extend the lifespan of a percentage of Sirt6-deficient mice (*Peshti et al., 2017*), all the analyses in our study have been done on littermate Sirt6$^{-/-}$Trp53$^{+/+}$ and Sirt6$^{-/-}$Trp53$^{+/-}$ mice (both on mixed backgrounds), where the Sirt6$^{-/-}$Trp53$^{+/+}$ mice had reduced lifespan of around 4 weeks and the Sirt6$^{-/-}$Trp53$^{+/-}$ mice exhibited extended maximal lifespan over the littermate Sirt6$^{-/-}$Trp53$^{+/+}$ mice (*Figure 2A–E*). This limits the possibility of mixed genetic background effects in our study. Moreover, the median lifespan for both male and

female Sirt6$^{-/-}$Trp53$^{+/-}$ mice on mixed background observed in our study is significantly longer than that of the reported Sirt6$^{-/-}$Trp53$^{+/+}$ mice on mixed background (*Peshti et al., 2017*). This suggests that the observed lifespan extension of compound mutant mice is indeed attributable largely, if not all, to *Trp53* heterozygosity. Moreover, *Peshti et al. (2017)* reported gender-specific differences in lifespan extension upon Sirt6 deficiency in mixed genetic background. Although we did observe that the maximal lifespan of the female Sirt6$^{-/-}$Trp53$^{+/-}$ mice was slightly more than the male Sirt6$^{-/-}$Trp53$^{+/-}$ mice (*Figure 2D and E*), the differences were not as striking as reported, suggesting further lifespan extension in Sirt6-deficient mice by *Trp53* haploinsufficiency reduces the percentage of gender-specific difference in lifespan. However, retinal disorders were prominently observed in our Sirt6$^{-/-}$Trp53$^{+/-}$ mice similar with that reported in *Peshti et al. (2017)*, consistent with a previous report stating that Sirt6 plays critical roles in the maintenance of retinal functions (*Silberman et al., 2014*). Another possibility cannot be eliminated that the very short lifespan of Sirt6 KO mice could result from oversensitivity to p53 activation associated with the genetic backgrounds used in the respective studies, thus resulting in enhanced cell death even at low DNA damage levels, thereby accelerating aging. Reducing *Trp53* levels by haploinsufficiency likely reduces this effect and hence delays aging, thereby allowing longer lifespan. Regarding cancer incidence, the types of tumors exhibited by Sirt6$^{-/-}$Trp53$^{+/-}$ and Sirt6$^{+/+}$Trp53$^{+/-}$ mice were very distinct. Given that Sirt6 has already been established as a potent tumor suppressor by several independent studies (*Lerrer and Cohen, 2013*; *Sebastián et al., 2012*), this explains as to why Sirt6$^{-/-}$Trp53$^{+/-}$ mice exhibit an earlier onset of tumorigenesis and mostly develop tumors other than sarcoma. Hence, sirt6 depletion enhanced the onset and diversified the type of tumors in the Trp53$^{+/-}$ background.

The dramatic lifespan extension observed in the mouse model reported in this study is significantly longer than any other rescue reported so far in mammalian aging models. The lifespan extension reported here is six times longer than the highest rescue of premature aging achieved so far in Sirt6-deficient mice (*Kawahara et al., 2009*). This suggests that the activated p53 pathway may be one of the most critical mechanisms underlying the accelerated aging and premature death in the absence of Sirt6. This further reinstates the importance of the SIRT6-p53 axis in cellular senescence and organismal aging. To also note here is that, though significant, the lifespan extension and phenotype amelioration by *Trp53* haploinsufficiency in Sirt6-deficient mice represent a partial rescue when compared with the wild-type mice. Given that Sirt6 regulates a range of pathways such as NF-κB, IGF1, AKT and others (*Dominy et al., 2012*; *Kanfi et al., 2012*; *Kawahara et al., 2009*; *Pan et al., 2016*; *Xiao et al., 2010*), it is tempting to speculate that these pathways may work in orchestra to contribute to the severe premature aging in mice in the absence of Sirt6. It is conceivable that Sirt6 null mice with *Trp53* haploinsufficiency can serve as a better model in evaluating the contribution of other signaling pathways to the premature aging in the absence of Sirt6. In addition, targeting multiple signaling pathways at the same time may further extend the lifespan in Sirt6-deficient mice. Given the recent report stating that inhibition of the interaction between FOXO4 and p53 rescues p53-mediated senescence and aging (*Baar et al., 2017*), it would be interesting to analyze if a similar administration of FOXO4 inhibitor could ameliorate aging-associated abnormalities in Sirt6-deficient mice.

Although the mammalian SIRT1 protein is also known to deacetylate p53 (*Langley et al., 2002*), the lack of any significant rescue of the abnormalities of Sirt1-deficient mice by loss of p53 (*Kamel et al., 2006*) clearly suggests, when viewed alongside our results, distinct roles for SIRT1 and for SIRT6 in the modulation of p53 activity and aging/longevity. In addition, the upregulation of p53 stability and activity in Sirt6-null scenario and the severe premature aging observed in Sirt6-deficient mice, clearly suggest the insufficiency of endogenous SIRT1 in compensating the hyperactivity of p53 under Sirt6-deficient conditions.

Apart from a better understanding of the contribution of different pathways regulated by SIRT6 to the aging process at the organismal level, the extended lifespan in Sirt6-deficient mice with *Trp53* haploinsufficiency also provides an opportunity to understand the in vivo role for Sirt6 in development, particularly during the maturation as the compound mutant mice survived remarkably long enough with growth retardation (*Figure 2*). On the other hand, given both SIRT6 and p53 are potent tumor suppressors (*Donehower and Lozano, 2009*; *Sebastián et al., 2012*) and their regulatory roles in aging, this study opens a new window for future study in the control of choice between aging and cancer development by the balance and cross-talk of p53 and SIRT6. p53 has been widely implicated in premature senescence and aging and is the focus of a large number of studies to

develop therapies. For example, independent studies have reported the efficacy of p53 inhibitors in promoting neuroprotection against age-associated neurodegenerative diseases (*Culmsee et al., 2001*; *Duan et al., 2002*; *Zhu et al., 2002*). Hence, identification of SIRT6 as a negative regulator of p53 unveils new avenues of research not only for basic scientists but also for those working to develop intervention and therapies against cancer and aging.

## Materials and methods

### Cell culture, transfection, recombinant proteins, reagent treatment and antibodies

HEK293 cells have been purchased from ATCC. HEK293 cells and mouse embryonic fibroblasts (MEFs) were cultured in DMEM supplemented with 10% fetal bovine serum (FBS) in 37°C incubators with 5% $CO_2$ and atmospheric oxygen conditions. HEK293 SIRT6 KO cells were a kind gift from Dr. Baohua Liu (Shenzhen University, China). Cell lines have been authenticated by short tandem repeat (STR) profile analysis and genotyping. Mycoplasm contamination were routinely examined by PCR. For analyzing effects of knocking down *Trp53* (alternatively denoted as p53), we generated $Sirt6^{-/-}Trp53^{+/-}$ compound mutant MEFs along with their WT ($Sirt6^{+/+}Trp53^{+/+}$) and Sirt6 KO ($Sirt6^{-/-}Trp53^{+/+}$) littermates at E12.5. Mycoplasma contamination was analyzed by DNA staining with DAPI. Transfection was performed with X-tremeGENE HP DNA Transfection Reagent (Roche, USA) and Lipofectamine 3000 (Invitrogen, USA). FLAG-tagged full length SIRT6 construct was obtained from addgene. Flag-tagged SIRT6 catalytically inactive mutant was kindly provided by Dr. Katrin Chua (Stanford school of Medicine, USA). FLAG-p53 and HA-P300 constructs were provided by Dr. Zhenkun Lou (Mayo clinic, USA). The point mutant constructs (p53 K381R, p53 K381Q) were generated using a site-directed mutagenesis kit (Agilent Technologies, QuikChange II XL). Antibodies against rabbit anti SIRT6, rabbit anti Acetyl p53 K373, rabbit anti Acetyl p53 K382 were obtained from Cell Signaling. Antibodies against rabbit anti Acetyl p53 K381 and rabbit anti H3 were purchased from Abcam. Rabbit anti H3K9ac, mouse anti γ-H2AX, and rabbit anti-PAN acetyl lysine antibodies were purchased from Millipore (Bedford, MA, USA). Rabbit anti H3K56ac antibody was purchased from Upstate. Anti p53 (DO-1), p53 (FL-393), and IgG antibodies were purchased from Santa Cruz (Santa Cruz, CA, USA). Mouse anti FLAG M2 antibody was purchased from Sigma. PE anti-mouse CD105 and FITC anti-mouse CD34 antibodies were purchased from eBiosciences. PE anti-mouse CD11b, APC anti-mouse CD44, PE-Cy5 anti-mouse CD8, and FITC anti-mouse CD4 antibodies were purchased from Biolegend. PE anti-mouse Sca-1, APC anti-mouse c-Kit, FITC anti-mouse CD31, and PerCP anti-mouse B220 antibodies were purchased from BD pharmingen. G agarose beads were purchased from Invitrogen, USA and anti-FLAG M2 affinity beads were purchased from Sigma. Cycloheximide (purchased from Sigma Aldrich) was used at a working concentration of 150 μg/ml. Recombinant Human SIRT6 protein was expressed in BL21 (DE3) strain using pET28a-sumo vector and was purified using Superdex200 gel-filtration column.

### In vitro SIRT6 deacetylation assay

SIRT6-mediated deacetylation was assayed on eluted p53 (overexpressed as a FLAG-tagged protein along with HA-tagged P300 and immunoprecipitated from HEK293 cells using FLAG antibodies, followed by elution with 3X FLAG peptide from Sigma) using the assay buffer as described in SIRT6 direct fluorescent screening assay kit by Cayman chemical (USA). Eluted acetyl p53 was incubated with recombinant human SIRT6 (rhSIRT6) for 45 min at 37°C, in the presence or absence of $NAD^+$ and nicotinamide. Acetylation of p53 was detected using antibodies against PAN-acetyl lysine (K), acetyl K381, acetyl K382 and acetyl K373.

### Generation of compound mutant mice

All animal works were performed with permission from local animal ethic committee (CULATR) and according to the guidelines and regulations. The $Sirt6^{co}$ floxed mutant mice have been purchased from Jackson laboratories and mated with β-actin-cre mice to generate $Sirt6^{+/-}Trp53^{+/+}$ mice. $Sirt6^{-/-}Trp53^{+/-}$ compound mutant mice were generated by compound heterozygous mating strategy since Sirt6 KO mice die within a month and p53 KO mice develop tumors very early and die prematurely. Briefly, $Sirt6^{+/-}Trp53^{+/+}$ and $Sirt6^{+/+}Trp53^{+/-}$ mice were bred to generate $Sirt6^{+/-}Trp53^{+/-}$

mice. Then Sirt6$^{+/-}$Trp53$^{+/-}$ mice were interbred to generate Sirt6$^{-/-}$Trp53$^{+/-}$ mice. Litters containing WT (Sirt6$^{+/+}$Trp53$^{+/+}$), Sirt6 KO (Sirt6$^{-/-}$Trp53$^{+/+}$) and compound mutant mice (Sirt6$^{-/-}$Trp53$^{+/-}$) were used for further analysis.

## Whole cell lysate collection, tissue sample collection, western blotting, and co-immunoprecipitation

Cells were harvested, washed with 1x PBS and resuspended in suspension buffer (0.1M NaCl, 10 mM Tris-HCl, pH 7.5, 1 mM EDTA, 1 mM DTT, pH 8.0, protease inhibitors, phosphatase inhibitors) followed by addition of an equal volume of laemmli buffer (0.1 M Tris-HCl, pH 7.0, 4% SDS, 20% glycerol, 1 mM DTT, protease inhibitors, phosphatase inhibitors) and boiled for 10 min. Tissue samples were minced and dounced thoroughly with suspension buffer, followed by addition of an equal volume of laemmli buffer and boiled immediately for 15 min. Western blotting was done as previously illustrated (*Liu et al., 2005*). Relative band intensity was measured by Image J and normalized to corresponding controls. Statistical analysis was performed using at least three independent immunoblots, which were quantified and two-tailed student's T test was used for calculating *P* values. Co-immunoprecipitation analyses were performed as described below. Briefly, cells were lysed with pre-chilled RIPA buffer containing 250 mM or 500 mM NaCl, protease inhibitors and phosphatase inhibitors. Primary antibodies or appropriate control IgGs were added to the lysates and incubated for 2 hr at 4°C on a rocking platform followed by addition of agarose beads and incubation was done O/N at 4°C. The beads were washed thrice with RIPA buffer (500 mM NaCl), resuspended with laemmli buffer and boiled for 10 min. The protein suspension was collected by centrifugation and stored at −80°C for western analysis.

## qPCR

Total RNA from cells or tissues (after douncing) was isolated using Trizol (Invitrogen) and 2 μl of the total extracted RNA was used for reverse-transcription reaction to generate cDNA using PrimeScript RT mastermix from Takara. Relative expression of the target genes was measured by qPCR and were normalized against respective Gapdh expression levels.

## Immunofluorescence staining and confocal microscopy

Cells were grown on chamber slides, fixed with 4% paraformaldehyde (in PBS), washed once in PBS, washed again with PBTr (PBS containing 0.1% Triton X-100) and blocked with 5% serum (FBS) in PBTr for 1 hr at RT (room temperature). Primary antibody was then diluted in PBTr and incubated O/N at 4°C. The slides were washed three times in PBTr, incubated with FITC-, TRITC-coupled secondary antibodies, or with Alexa-fluor 488/562 (donkey anti-rabbit), diluted in PBTr for 60–75 min at RT, washed three times with PBTr, then two times with PBS followed by mounting with *SlowFade*Â Gold antifade reagent with DAPI (Invitrogen, USA), sealed with nail polish and subjected for confocal microscopic analysis at room temperature. Images were obtained using 63X, 1.4 NA oil objective (Carl Zeiss LSM 700 inverted confocal microscope equipped with ZEN 2010 software version 6.0.0.309) with 405 nm, 488 nm and 555 nm laser illumination (standard excitation and emission filter sets). Images were processed using ZEISS ZEN lite software.

## Senescence-associated β-galactosidase staining

Senescence-associated β-galactosidase staining kit from Cell Signaling was used. Tissues were fixed with gelatin and cryopreserved before cutting sections. These cryosections were then immersed in 1x PBS and incubated at 37°C for 20–30 min, followed by a wash with 1x PBS. Then the slides were incubated with fixative solution for 10–15 min, washed one with 1x PBS, and incubated with β-galactosidase staining solution for O/N at 37°C in dark. The slides were then washed once with 1x PBS and mounted with *SlowFade*Â Gold antifade reagent (Invitrogen). For cells, 10$^5$ cells were plated onto six well plates, grown O/N in 37°C incubator, followed by aspiration of growth medium and washing twice with 1x PBS. The rest of the procedures were similar to that of staining of slides as mentioned above.

## Serum glucose and IGF-1 assay

Serum glucose and IGF-1 levels in mice were assayed using kits from Abcam and R and D Systems respectively, following the manufacturer's protocols.

## Immunostaining for FACS analysis

$10^7$ bone marrow cells were resuspended in 1x RBC lysis solution and kept on ice for 5 min, followed by centrifugation at 500 g for 5 min and washing with ice cold 1x PBS. The bone marrow stromal cells were then stained with 0.5 µl each of PE-Sca1, APC-cKit and FITC-CD34 for hematopoietic stem cell profile analysis and 0.5 µl each of PE-CD105, APC-CD44 and FITC-CD31 for mesenchymal stem cell analysis. Only Sca-1$^+$c-Kit$^+$CD34$^-$ cells were counted as hematopoietic stem cells and only CD105$^+$CD44$^+$CD31$^-$ cells were counted as mesenchymal stem cells in flow cytometric analysis. Mono-stained cells and cells stained with isotype controls were also analyzed simultaneously. Staining was done for 20–30 min on ice in dark, followed by washing with pre-chilled 1x PBS, and resuspension in 1x PBS and then taken for FACS analysis using BD FACSCantoII Analyzer. Similarly, thymic cells were stained with 0.5 µl of FITC-CD4 and PE/Cy5-CD8, keeping mono-stained and isotype control stained cells following the protocol mentioned above. Also, splenic cells were stained with 0.5 µl of PerCP-B220 and PE-CD11b separately keeping isotype controls, and similar procedures were followed to prepare for FACS analysis.

## Annexin-V staining

Analysis of apoptosis was done using TACS Annexin V-Biotin Apoptosis Detection kit from R and D systems. Cells from thymus and bone marrow stromal cells from tibia and femur of mice were flushed out using ice-cold 1x PBS. $10^6$ cells were washed once with ice cold 1x PBS, stained with propidium iodide and Annexin-V Biotin for 15 min in dark at RT, followed by centrifugation at 500 g for 5 min. Then the cells were incubated with FITC labelled Streptavidin antibody (BD Pharmingen, BD Biosciences, San Jose, USA) for 15 min in dark at RT, followed by centrifugation at 500 g for 5 min, resuspension in pre-chilled 1x PBS and flow cytometric analysis.

## Isolation of bone marrow stromal cells, thymic and splenic cells

After sacrificing the mice by cervical dislocation, both femurs and tibias were dissected, their ends were cut with a sharp and sterile scissor, and the bone marrow stromal cells were flushed out with pre-chilled 1x PBS using a 23$^{1/2}$ needle and were passed through 70-µm cell strainer for filtering out the clustered cells. The viable cell number was then calculated using hemocytometer after dilution with Trypan blue. Thymic and splenic cells were also isolated by flushing the thymus and spleen of mice using pre-chilled 1x PBS and cell number was calculated in a similar way.

## MTT assay

Measurement of cell viability was performed using TACS Cell Proliferation Assay Kit (R and D systems). Briefly, around 5000 MEF cells were seeded onto 96 well plates in triplicates and grown for 48 hr. 10 µl of MTT reagent was added to each well, keeping triplicate controls of non-treated blank sets for each cell type. The cells were placed back in 37°C incubator for 4 hr, and development of purple coloration was monitored. Then 100 µl of detergent reagent was added to each well, and the plate was left covered in dark for 2–4 hr, followed by measurement of absorbance at 570 nm in a microplate reader. Average values of blanks were subtracted from average values from triplicate readings and graph was plotted to analyze the same.

## X-ray imaging

X-ray imaging of the whole body of mice was done using UltraFocus1000 (fully-shielded X-ray cabinet) from Faxitron.

## Statistical analysis

Quantifications have been represented as mean ± SEM, n >= 3 for all experiments performed. Statistical significance was analyzed by two-tailed unpaired Student's $t$-test and one-way ANOVA using GraphPad Prism software. Kaplan-Meier Survival curves have been plotted using GraphPad Prism software, and Log-rank (Mantel-Cox) test has been employed to calculate statistical significance.

Since our sample sizes varied from 3 to 6, it was difficult to conclude whether normal distribution was achieved or not. Given that parametric tests are more sensitive and powerful in determining statistically significant differences which non-parametric tests might overlook, we either used Student's t-test or ANOVA to calculate statistical significance in our results.

## Acknowledgements

The authors want to thank Dr. Katrin Chua, Stanford university medical center, for catalytically inactive SIRT6 and GFP-tagged SIRT6 constructs. FLAG-p53 and HA-p300 constructs were a kind gift from Dr. Zhenkun Lou, Mayo clinic, USA. This work is supported by grants from NSFC (81330009, 81671399) Research Grant Council of Hong Kong (17123816 and HKU2/CRF/13G) and HMRF (03142456). The authors declare no competing financial and non-financial interests.

## Additional information

### Funding

| Funder | Grant reference number | Author |
|---|---|---|
| Research Grants Council, University Grants Committee | 17123816; HKU2/CRF/13G | Quan Hao<br>Zhongjun Zhou |
| National Natural Science Foundation of China | 81671399 | Xinguang Liu |
| National Natural Science Foundation of China | 81330009 | Zhongjun Zhou |
| Health and Medical Research Fund | 03142456 | Zhongjun Zhou |

The funders had no role in study design, data collection and interpretation, or the decision to submit the work for publication.

### Author contributions

Shrestha Ghosh, Designed and performed the experiments, Data analysis, Drafted the manuscript; Sheung Kin Wong, Investigation, Methodology, Writing—original draft; Zhixin Jiang, Investigation; Baohua Liu, Methodology; Yi Wang, Quan Hao, Vera Gorbunova, Resources; Xinguang Liu, Formal analysis; Zhongjun Zhou, Conceptualization, Supervision, Funding acquisition, Investigation, Writing—review and editing

### Author ORCIDs

Baohua Liu ORCID http://orcid.org/0000-0002-1599-8059
Zhongjun Zhou ORCID https://orcid.org/0000-0001-7092-8128

### Ethics

Animal experimentation: This study was performed in strict accordance with the recommendations in the Guide for the Care and Use of Laboratory Animals of the National Institutes of Health. All of the animals were handled according to approved CULATR protocols (3575-15) of the University of Hong Kong. The protocol was approved by the Committee on the Ethics of Animal Experiments of the University of Hong Kong (Permit Number: 3575-15). All surgery was performed under sodium pentobarbital anesthesia, and every effort was made to minimize suffering.

### Decision letter and Author response

Decision letter https://doi.org/10.7554/eLife.32127.021
Author response https://doi.org/10.7554/eLife.32127.022

## Additional files

**Supplementary files**

• Transparent reporting form
DOI: https://doi.org/10.7554/eLife.32127.019

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
