## [Decision Letter]

Thank you for submitting your article "Haploinsufficiency of *p53* dramatically extends the lifespan of Sirt6-deficient mice" for consideration by *eLife*. Your article has been reviewed by four peer reviewers, and the evaluation has been overseen by Reinhard Fässler as Reviewing Editor and Jessica Tyler as the Senior Editor. The following individuals involved in review of your submission have agreed to reveal their identity: Paul Hasty (Reviewer #1); George Garinis (Reviewer #3).

The reviewers have discussed the reviews with one another and the Reviewing Editor has drafted this decision to help you prepare a revised submission.

Summary:

The manuscript reports that heterozygous deletion of p53 dramatically extends lifespan of Sirt6-deficient mice and improves a number of aging-associated symptoms, including kyphosis, bone marrow stem cell decline, colitis and reduces premature senescence and apoptosis. At the molecular level SIRT6 deacetylates p53 to negatively regulate stability and activity of p53. These findings demonstrate that SIRT6 acts on p53, thereby preventing several features of accelerated aging.

Essential revisions:

1) What was their cancer incidence? Since there is significant overlap between the life span curves for sirt6^-/-^ p53^+/-^ mice and the Sirt6^+/+^ p53^+/-^ it would be interesting to know if Sirt6 deletion reduces cancer incidence in the p53^+/-^ background.

2) Genetic background of mouse strains is notorious in terms of clinical phenotypes. The authors have used a F2 of pure FVB and pure C57Bl6, which is a wise strategy as it reduces risks associated with the use of only one genetic background which may have unanticipated strain-specific effects on the phenotype. This is even more important as recently the dramatic effect of genetic background on lifespan of Sirt6 mutants has become apparent from several studies including Peshti et al., (2017), using a mixed 129/SvJ/BALB/c background, which extended lifespan in females over 10-fold and in males approximately 5-fold. Also, several aging features exhibited by Sirt6 mutant mice have been found to vary significantly with genetic background. In fact, one explanation for the very short lifespan of the Sirt6 KO mice in the current and other reports could well be that these genetic backgrounds are associated with oversensitivity to p53 activation, causing cell death already at low DNA damage levels thereby accelerating aging. Reducing p53 levels by haplo-insufficiency would reduce this effect and hence delay aging and allow longer lifespan. This possibility should be considered.

3) It seems that what p53 haploinsufficiency rescues in Sirt6 mice is mostly the growth defect seen in Sirt6-/- animals. This is clearly a postnatal developmental defect that likely reflects the GH/IGF1 axis (in the liver) of these animals and has been shown (also in Sirt6 mice) in other mouse models carrying e.g. DNA repair defects in Nucleotide Excision Repair. The manuscript would benefit from some insights/data on this pathway in Sirt6^-/-^ and Sirt6^-/-^p53^+/-^ mice as it would help putting their working model into a larger context. Any data on the metabolic status of Sirt6^-/-^p53^+/-^ animals would also be useful (glucose, triglyceride levels in the serum of Sirt6^-/-^p53^+/-^ mice). The authors provide no information on the frequency of spontaneous tumors in the longest-lived Sirt6^-/-^p53^+/-^ animals. If such data are available, it would be relevant to include them in this work.

4) The MEF experiments seem carefully done, using independent triplicates and only early passage cells. However, MEFs are extremely unstable and may already acquire chromosomal abnormalities in the first passage of culturing at atmospheric oxygen. Therefore, it is important that MEFs are grown at low (3%, i.e. physiological) levels of O2. This is even more paramount for the p53 and Sirt6 mutants used here because of compromised genome stability mechanisms. M&M does not indicate the conditions under which the MEFs were grown. This point should be clarified in the manuscript.

5) Can the authors provide information on the cause of death of the Sirt6^-/-^p53wt mice, which live only 4 weeks. In addition, what is the cause of death of the Sirt6^-/-^p53^+/-^ mice? Were causes of death upon haploinsufficiency of p53 different from those with wt p53?

6) Given the reported role of SIRT6 in DNA repair it is remarkable that no increased S15-Phosphorylation of p53 was observed in Sirt6^-/-^ MEFs (Figure 6 and subsection “SIRT6 negatively regulates the stability of p53”), since one would expect that spontaneous DNA damage levels would be elevated in the absence of SIRT6. What is the explanation?

7) The authors use Students unpaired t-test throughout for statistics. This seems inappropriate on occasions where multiple experimental groups are compared, where an ANOVA-type of test would be more appropriate (for example all bar graph quantifications in Figure 1 and Figure 2). Also, throughout, the authors should indicate how normal distribution of the data was confirmed and in cases where no normal distribution is found, a non-parateric test should be used.

8) It would be also more informative to plot individual data points in the graphs, instead of using bar graphs with SEM.

---

## [Author Response]

Essential revisions:1) What was their cancer incidence? Since there is significant overlap between the life span curves for sirt6^-/-^ p53^+/-^ mice and the Sirt6^+/+^ p53^+/-^ it would be interesting to know if Sirt6 deletion reduces cancer incidence in the p53+/- background.

We thank the reviewers for their insightful questions and comments. Regarding cancer incidence, some of the *Sirt6^+/+^p53^+/+^* (WT) mice (approximately 6-8%) developed tumors at around 2-2.5 years, as expected. Approximately 26-28% of *Sirt6^+/+^p53^+/-^* mice developed tumors by 17-20 months, regardless of gender. This is in consistence with the existing literature stating that haploinsufficiency of *p53* in wild-type background results in increased cancer incidence (Jacks et al., 1994; Tyner et al., 2002). The *Sirt6^-/-^p53^+/+^* mice mostly died within 4 weeks of birth, and no incidence of cancer was observed in this genotype. Interestingly, around 32% of the *Sirt6^-/-^p53^+/-^* (compound mutant) mice were detected with obvious tumors upon necropsy close to their terminal lifespan. However, the incidence of cancer in *Sirt6^-/-^p53^+/-^* mice was earlier (9-16 months) than that of the *Sirt6^+/+^p53^+/-^* mice (17-20 months).

The types of tumors exhibited by *Sirt6^-/-^p53^+/-^*and *Sirt6^+/+^p53^+/-^* mice were very distinct. In consistence with existing literature (Jacks et al., 1994; Tyner et al., 2002), the *Sirt6^+/+^p53^+/-^* mice mostly developed sarcoma by 17-20 months (Figure 3—figure supplement 1). However, the *Sirt6^-/-^p53^+/-^* mice developed other types of tumors, such as kidney tumor (19%), prostate tumor (17%), bladder tumor (16%), and pancreatic tumor (8%) (Figure 3—figure supplement 1 respectively), while only 3% of them exhibited incidence of sarcoma. Given that Sirt6 has already been established as a potent tumor suppressor by several independent studies (Sebastian et al., 2012; Lerrer and Cohen, 2013), this explains as to why *Sirt6^-/-^p53^+/-^* mice exhibit an earlier onset of tumorigenesis and mostly develop tumors other than sarcoma. Indeed, sarcoma incidence in the compound mutants is reduced compared with *p53*^+/-^ mice. Hence, although *Sirt6^-/-^p53^+/-^*and *Sirt6^+/+^p53^+/-^* mice showed overlap between their lifespan curves, Sirt6 depletion did not reduce, but rather enhanced the onset and diversified the type of tumors in the *p53^+/-^* background, given that Sirt6 is a tumor suppressor itself. These discussions have now been included in the Results section, the Discussion section, and the data on cancer incidence in *Sirt6^-/-^p53^+/-^*and *Sirt6^+/+^p53^+/-^* mice have been presented in Figure 3—figure supplement 1.

2) Genetic background of mouse strains is notorious in terms of clinical phenotypes. The authors have used a F2 of pure FVB and pure C57Bl6, which is a wise strategy as it reduces risks associated with the use of only one genetic background which may have unanticipated strain-specific effects on the phenotype. This is even more important as recently the dramatic effect of genetic background on lifespan of Sirt6 mutants has become apparent from several studies including Peshti et al., (2017), using a mixed 129/SvJ/BALB/c background, which extended lifespan in females over 10-fold and in males approximately 5-fold. Also, several aging features exhibited by Sirt6 mutant mice have been found to vary significantly with genetic background. In fact, one explanation for the very short lifespan of the Sirt6 KO mice in the current and other reports could well be that these genetic backgrounds are associated with oversensitivity to p53 activation, causing cell death already at low DNA damage levels thereby accelerating aging. Reducing p53 levels by haplo-insufficiency would reduce this effect and hence delay aging and allow longer lifespan. This possibility should be considered.

We are thankful to the reviewers for their observation and discussion of the results. We have now incorporated this discussion presented by the reviewers in our Discussion section.

3) It seems that what p53 haploinsufficiency rescues in Sirt6 mice is mostly the growth defect seen in Sirt6-/- animals. This is clearly a postnatal developmental defect that likely reflects the GH/IGF1 axis (in the liver) of these animals and has been shown (also in Sirt6 mice) in other mouse models carrying e.g. DNA repair defects in Nucleotide Excision Repair. The manuscript would benefit from some insights/data on this pathway in Sirt6-/- and Sirt6-/-p53+/- mice as it would help putting their working model into a larger context. Any data on the metabolic status of Sirt6-/-p53+/- animals would also be useful (glucose, triglyceride levels in the serum of Sirt6-/-p53+/- mice). The authors provide no information on the frequency of spontaneous tumors in the longest-lived Sirt6-/-p53+/- animals. If such data are available, it would be relevant to include them in this work.

We thank the reviewers for their comments. To this end, we analyzed the GH/IGF1 axis in the mice by checking their serum IGF1 levels. As previously reported (Mostoslavsky et al., 2006), the serum IGF1 levels were drastically reduced in *Sirt6^-/-^p53^+/+^* mice as compared to *Sirt6^+/+^p53^+/+^* mice at the age of approximately 24 days (Figure 2—figure supplement 1). This is reminiscent of the notable reduction of serum IGF1 levels in other mouse models carrying DNA repair defects (Niedernhofer et al., 2006). Interestingly, although less than that of wild-type littermates, the serum IGF1 levels in *Sirt6^-/-^p53^+/-^*mice were significantly rescued with respect to *Sirt6^-/-^p53^+/+^*mice at the age of around 24 days (Figure 2—figure supplement 1). The serum IGF1 levels of *Sirt6^-/-^p53^+/-^*mice were slightly lower than that of their wild-type littermates at their terminal stage (10-16 months) (Figure 2—figure supplement 1). However, the values did not reach significance. This suggests that *p53* haploinsufficiency likely rescues the abnormal IGF-1 levels in *Sirt6^-/-^p53^+/+^* mice, thus contributing to the rescue in growth defects and hence lifespan of the mutant mice. However, death of the compound mutant mice is unlikely due to the reduction of IGF1. The data on serum IGF1 levels have now been included in the Results section and presented in Figure 2—figure supplement 1.

Apart from serum IGF1 levels, we also analyzed serum glucose levels in the mice. Although *Sirt6^-/-^p53^+/+^* mice displayed a decline in their serum glucose levels (Figure 2—figure supplement 1), it was not a very significant drop. However, the compound mutant (*Sirt6^-/-^p53^+/-^*) mice exhibited an almost comparable serum glucose level to their wild-type littermates at the age of 24 days (Figure 2—figure supplement 1). We also analyzed the serum glucose levels of *Sirt6^-/-^p53^+/-^*mice during their terminal lifespan (i.e. at 10-16 months of age). Interestingly, *Sirt6^-/-^p53^+/-^*mice had slightly reduced serum glucose levels than their wild-type littermates (Figure 2—figure supplement 1), thus suggesting that severe metabolic disorders may not be the cause of death for *Sirt6^-/-^p53^+/-^*mice. The data on serum glucose levels have now been included in the Results section, and presented in Figure 2—figure supplement 1.

Regarding cancer incidence, around 32% of the *Sirt6^-/-^p53^+/-^* (compound mutant) mice developed tumors during their terminal lifespan (i.e. around 9-16 months), while the *Sirt6^+/+^p53^+/-^* mice mostly developed sarcoma by 17-20 months. The *Sirt6^-/-^p53^+/-^* mice developed other types of tumors, such as kidney tumor (19%), prostate tumor (17%), bladder tumor (16%), and pancreatic tumor (8%), while only 3% of them exhibited incidence of sarcoma. Given that Sirt6 has already been established as a potent tumor suppressor by several independent studies (Sebastian et al., 2012; Lerrer and Cohen, 2013), this explains as to why *Sirt6^-/-^p53^+/-^* mice exhibited an earlier onset of tumorigenesis than *Sirt6^+/+^p53^+/-^* mice and mostly developed tumors other than sarcoma. The findings of cancer incidence have been discussed in detail in point 1. The data on cancer incidence have been presented in Figure 3—figure supplement 1B-F and have been discussed in the Results section and Discussion section in the manuscript.

4) The MEF experiments seem carefully done, using independent triplicates and only early passage cells. However, MEFs are extremely unstable and may already acquire chromosomal abnormalities in the first passage of culturing at atmospheric oxygen. Therefore, it is important that MEFs are grown at low (3%, i.e. physiological) levels of O2. This is even more paramount for the p53 and Sirt6 mutants used here because of compromised genome stability mechanisms. M&M does not indicate the conditions under which the MEFs were grown. This point should be clarified in the manuscript.

We thank the reviewers for their suggestion. We cultured the early passage MEF cells (P0, P1 and P2) under low (3%) levels of O2. Although the *Sirt6^+/+^p53^+/+^* and *Sirt6^-/-^p53^+/-^*MEFs thrived well initially, the *Sirt6^-/-^p53^+/+^*MEFs were mostly dead in the culture and could not be passaged further. The cell death of P2 *Sirt6^-/-^p53^+/+^*MEFs at 3% O2 conditions could not even be averted after they have been cultured well at P1 under atmospheric oxygen conditions. *Sirt6^-/-^p53^+/-^*MEFs exhibited increased cell survival at P2 and P3 compared with that of *Sirt6^-/-^p53^+/+^*MEFs. Several independent studies have reported that Sirt6 subdues the damage incurred due to hypoxic conditions in various cells lines and tissues (Maksin-Matveev et al., 2015; Kok et al., 2015; Shun et al., 2016; Lee et al., 2016). Loss of Sirt6 in MEFs makes them more vulnerable to cell death under low oxygen conditions. We have now included the culture conditions for MEF cells used in our study in the Materials and methods section.

5) Can the authors provide information on the cause of death of the Sirt6-/-p53wt mice, which live only 4 weeks. In addition, what is the cause of death of the Sirt6-/-p53+/- mice? Were causes of death upon haploinsufficiency of p53 different from those with wt p53?

*Sirt6^-/-^p53^+/+^* mice have been reported to die because of acute aging-associated degenerative processes, such as severe organ degeneration, body wasting, abruptly lowered serum IGF-1 levels, chronic metabolic disorders and other degenerative aging-associated phenotypes (Mostoslavsky et al., 2006). It is likely that gut inflammation also contributes to the early death of the animals (Mostoslavsky et al., 2006). Although the *Sirt6^-/-^p53^+/-^* mice always weighed less than their wild-type littermates (Figure 2), the *Sirt6^-/-^p53^+/-^* mice showed signs of severe body wasting during their terminal lifespan (around 10-16 months). A major fraction of both female and male *Sirt6^-/-^p53^+/-^*mice developed severe degenerative phenotypes, with acute loss of vigour and numbness of limbs during their terminal lifespan, and we had to sacrifice those mice when they reached humane end-point at the age of 14-16 months. Upon necropsy, around 32% of the *Sirt6^-/-^p53^+/-^* (compound mutant) mice exhibited tumors during their terminal lifespan stage (around 13-16 months for females and around 9-14 months for males). Some *Sirt6^-/-^p53^+/-^*female mice succumbed to the disorder of incontinence at around 14-16 months, while around 95% of *Sirt6^-/-^p53^+/-^*male mice exhibited penile protrusion at around 9-11 months of age (now included as Figure 3—figure supplement 3F). However, gross metabolic disorders were not evident in most of the terminally ill *Sirt6^-/-^p53^+/-^*mice, as evident from their slightly reduced serum glucose levels as compared to their wild-type littermates (Figure 2—figure supplement 1). Taken together, the *Sirt6^-/-^p53^+/-^* mice died of disorders which are both overlapping with and distinct from *Sirt6^-/-^p53^+/+^* mice. The causes of death of *Sirt6^-/-^p53^+/-^* mice have been included in the Results section and Discussion section.

6) Given the reported role of SIRT6 in DNA repair it is remarkable that no increased S15-Phosphorylation of p53 was observed in Sirt6-/- MEFs (Figure 6 and subsection “SIRT6 negatively regulates the stability of p53”), since one would expect that spontaneous DNA damage levels would be elevated in the absence of SIRT6. What is the explanation?

Although SIRT6 plays a critical role in DNA damage repair, it has been reported by several independent studies that the endogenous levels of DNA damage in Sirt6-deficient cells are not prominently detected without any exogenous DNA damage induction. For example, enhanced γ-H2AX levels are not evident in SIRT6-deficient cells without any external DNA damage induction (Mostoslavsky et al., 2006; Kaidi et al., 2010; Toiber et al., 2013). Hence, the inherent DNA damage levels are likely not intense enough to be detected by molecular assays, such as analysis of H2AX levels or p53 phosphorylation levels by Western blotting or immunofluorescence staining. It is possible that other repair mechanisms may compensate for the damage incurred from inherent DNA damage in the absence of SIRT6. This might explain why S15 phosphorylation levels of p53 are not significantly elevated in SIRT6-deficient cells without exogenous induction of DNA damage.

7) The authors use Students unpaired t-test throughout for statistics. This seems inappropriate on occasions where multiple experimental groups are compared, where an ANOVA-type of test would be more appropriate (for example all bar graph quantifications in Figure 1, Figure 2). Also, throughout, the authors should indicate how normal distribution of the data was confirmed and in cases where no normal distribution is found, a non-parateric test should be used.

We thank the reviewers for their comments. We have now used one-way ANOVA to analyze statistical significance in all quantifications in Figure 1 and Figure 2, except Figure 1, where only two groups have been compared for statistical significance using Student’s t-test. We have also used one-way ANOVA to calculate statistical significance in all quantifications in Figure 4, Figure 5—figure supplement 1 and Figure 5—figure supplement 1 because they involved comparison within multiple groups.

Regarding the normal distribution of data, it is hard to determine normality with small sample sizes, which in our case ranges from 3-6. Hence it became difficult for us to analyze how skewed our data is, thus making it tough to conclude whether normal distribution has been achieved or not. In such a scenario, since parametric tests are more sensitive and powerful in determining statistically significant differences which non-parametric tests might overlook, we either used Student’s t-test or ANOVA to calculate statistical significance in our results. However, for Figure 2, where our sample sizes are more than 30, we realized that normal distribution was not reached. Hence, we used Log-Rank (Mantel-Cox) test, which is a non-parametric test, to calculate statistical significance for the Kaplan Meier survival curves of the mice (Figure 2). All parametric and non-parametric tests performed in our study have been analyzed using GraphPad Prism. The Results section and all Figure legends have been accordingly written, and the above discussion on parametric, non-parametric tests and employment of ANOVA has been mentioned in the Materials and methods section.

8) It would be also more informative to plot individual data points in the graphs, instead of using bar graphs with SEM.

We thank the reviewers for their suggestion. For serum glucose and IGF1 levels in mice, the individual data points have been plotted in the graphs presented in Figure 5—figure supplement 1. Apart from these, the quantifications depicting relative intensity of Western blotting data or relative expression of genes via qPCR have been shown with bar graphs representing mean values of individual data points within a group with error bars showing SEM. We have now described our graph values more clearly in the Figure legends.